# Stabilizing the Kumaraswamy Distribution

**Max Wasserman**                                        *mwasser6@cs.rochester.edu*
*Department of Computer Science*
*University of Rochester*

**Gonzalo Mateos**                                        *gmateosb@ece.rochester.edu*
*Department of Electrical and Computer Engineering*
*University of Rochester*

**Reviewed on OpenReview:** *https://openreview.net/forum?id=XXXX*

## Abstract

Large-scale latent variable models require expressive continuous distributions that support efficient sampling and low-variance differentiation, achievable through the reparameterization trick. The Kumaraswamy (KS) distribution is both expressive and supports the reparameterization trick with a simple closed-form inverse CDF. Yet, its adoption remains limited. We identify and resolve numerical instabilities in the log-pdf, CDF, and inverse CDF, exposing issues in libraries like PyTorch and TensorFlow. We then introduce simple and scalable latent variable models to address exploration-exploitation trade-offs in contextual multi-armed bandits and facilitate uncertainty quantification for link prediction with graph neural networks. We find these models to be most performant when paired with the stable KS. Our results support the stabilized KS distribution as a core component in scalable variational models for bounded latent variables.

## 1 Introduction

Probabilistic models use probability distributions as building blocks to model complex joint distributions between random variables. Such distributions can model unobserved 'latent' variables $z$, or observed 'data' variables $x$. Bounded interval-supported latent variables are central to many key applications, such as unobserved probabilities (e.g., user clicks in recommendation systems or links between network nodes), missing measurements in control systems (e.g., joint angles in $[0, 2\pi]$), and stochastic policies over bounded actions in reinforcement learning (e.g., motor torque in $[-10, 10]$).

To meet the demands of large-scale latent variable models, distributions supported on bounded intervals must satisfy the following criteria: (i) support the reparameterization trick through an explicit reparameterization function, such as a closed-form inverse CDF, enabling low-variance gradient estimation and efficient sampling; (ii) provide sufficient expressiveness to capture complex latent spaces; and (iii) offer simple distribution-related functions (log-pdf, explicit reparameterization function, and gradients) that allow fast and accurate evaluation. In Section 2, we argue that the Kumaraswamy (KS) distribution uniquely meets these criteria, yet remains surprisingly underused. In Section 3, we demonstrate that the KS distribution-related functions exhibit numerical instabilities concealed by standard parameterizations and exacerbated in large-scale latent variable models.

In this paper, we make the following technical contributions:

- We introduce an unconstrained logarithmic parameterization of the KS's log-pdf, CDF, inverse CDF, and gradients, which isolate the dominant numerical instabilities, allowing application of recently developed stabilization techniques (Section 3).

- We illustrate how the KS can be used as an integral component of latent variable models and demonstrate its benefits over alternative distributions. In addition to the workhorse variational autoencoder (VAE),

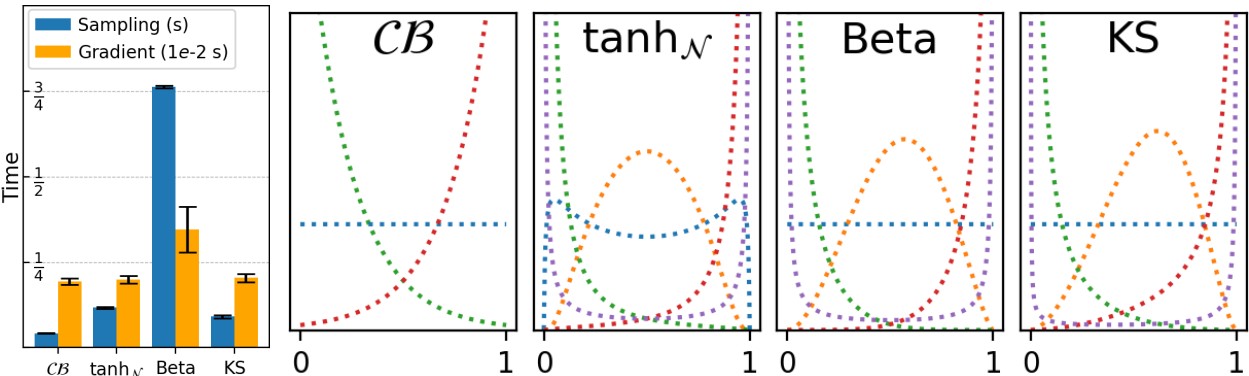

Figure 1: Comparison of relevant bounded interval-supported distributions. *Left:* Time for sampling and differentiating through samples. The Beta lacks explicit reparameterization, and has slower sampling and gradients. *Right:* Expressiveness in terms of attainable prototypical shapes.

our experiments showcase: (i) the Variational Bandit Encoder (VBE), addressing exploration-exploitation trade-offs in contextual Bernoulli multi-armed bandits (Section 4.2); and (ii) the Variational Edge Encoder (VEE), facilitating uncertainty quantification in link prediction with graph neural networks (Section 4.3).

The VBE and VEE are scalable latent variable models with bounded interval-supported latent variables. Unlike traditional methods which tend to model global latent variables (Section 5), such as the parameters of a shared neural network (NN), the VBE and VEE define local latent variables per bandit arm or network link. This allows the models to incorporate prior knowledge precisely where domain expertise tends to reside—at a granular level, such as the expected reward of a specific arm or the probability of a particular link's existence. Our numerical experiments demonstrate that both models tend to perform best when paired with the stabilized KS distribution in their variational posterior, reinforcing its role as a core component in large-scale bounded latent variable modeling. Our experiments in Sections 4.2-4.3 provide evidence toward the benefits of adopting the KS in timely application domains — enabled by our stable parameterization — without making claims on improvements over state-of-the-art models, which future work may investigate.

## 2 Background

The KS distribution (Jones, 2009; Kumaraswamy, 1980) has pdf $f(x) = abx^{a-1}(1-x^a)^{b-1}$, CDF $F(x) = 1 - (1-x^a)^b$, and inverse CDF $F^{-1}(u) = (1-u^{b^{-1}})^{a^{-1}}$, all defined for $x, u \in (0,1)$ and parameterized by $a, b > 0$. The differential entropy of a KS with parameters $a, b$ is

$$\mathcal{H}(\text{KS}) := -\int_0^1 f(x) \log f(x) dx$$
$$= 1 - b + (1-a)\left(\phi^{(0)}\left(b^{-1}+1\right)+\gamma\right) - \log a - \log b,$$

where $\phi^{(0)}$ is the digamma function and $\gamma \approx 0.577$ is the Euler-Mascheroni constant. The digamma function and its gradient, the trigamma function $\phi^{(1)}(x)$, can be represented as infinite series which converge rapidly and thus can be used effectively in numerical applications. They are included as standard functions in common auto-differentiation frameworks.

**Continuous distributions with bounded interval support.** Among distributions with bounded interval support, the KS uniquely satisfies desiderata (i)–(iii) in Section 1. It supports the reparameterization trick through its closed-form, differentiable inverse CDF, providing efficient sampling and low-variance gradients. The KS supports four distinct prototypical shapes — bell, U, increasing, and decreasing (Figure 1, right) — providing expressivity for diverse modeling tasks. Its log-pdf, CDF, and inverse CDF, along with their gradients, are composed only of affine transformations, exponentials, and logarithms, and can be

| Property / Distribs. | $\mathcal{CB}$ | $\tanh_{\mathcal{N}}$ | Beta | KS |
|---|---|---|---|---|
| Relative Expressiveness | low | high | high | high |
| Gradient Reparam. | explicit | explicit | implicit | explicit |
| Contains Uniform | ✓ | ✗ | ✓ | ✓ |
| Closed-form CDF | ✓ | ✗ | ✗ | ✓ |
| Closed-form inverse CDF | ✓ | ✗ | ✗ | ✓ |
| Numerical Issues | mild | high | low | low |
| Complex Functions | $\tanh^{-1}$ | $\log(1\text{-}\tanh^2(x))$ | $\beta, I$ | None |
| Parameterization | $\mathbb{R}$ | $\mathbb{R}^2$ | $\mathbb{R}^2_+$ | $\mathbb{R}^2$ |
| Analytical Moments | ✓ | ✗ | ✓ | ✓ |
| Closed-form KL | Exp. Family | $\tanh_{\mathcal{N}}$ | Exp. Family | Beta |
| Entropy $\mathcal{H}$ | ✓ | ✗ | ✓ | ✓ |

Table 1: Comparison of bounded interval-supported distribution families.

parameterized directly in terms of unconstrained logarithmic values; see Section 3. This enables straightforward implementation with minimal dependencies and keeps most computation in log-space, enhancing stability and accuracy. The unconstrained logarithmic parameterization makes it well-suited for NNs, eliminating the need for positivity-enforcing link functions. Additionally, the KS has differentiable, closed-form expressions for moments, median, differential entropy $\mathcal{H}(\text{KS})$, and the Kullback-Leibler (KL) divergence to the Beta distribution, facilitating efficient incorporation of prior information.

We briefly introduce workhorse bounded-interval supported distribution families, namely the the Continuous Bernoulli, the Beta, and the tanh-squashed-Gaussian. The Continuous Bernoulli ($\mathcal{CB}$) (Loaiza-Ganem & Cunningham, 2019) arises in deep learning for modeling continuous $[0, 1]$-valued pixel intensities in natural images. It provides a normalized probabilistic counterpart to the commonly used binary cross-entropy loss, with density $p(x; \lambda) = C(\lambda)\lambda^x(1 - \lambda)^{1-x}, x \in [0, 1], \lambda \in (0, 1)$, where $C(\lambda) = \{2$ if $\lambda = \frac{1}{2}$, else $\frac{2\tanh^{-1}(1-2\lambda)}{1-2\lambda}\}$ is the normalizing constant. The Beta distribution is a flexible two-parameter family, widely used for modeling probabilities and proportions. Its density, parameterized by $a, b > 0$, is given by: $p(x; a, b) = B(a, b)^{-1}x^{a-1}(1 - x)^{b-1}, x \in (0, 1)$, where $B(a, b)$ is the Beta function. The tanh-squashed-Gaussian ($\tanh_{\mathcal{N}}$) maps Gaussian samples through the tanh function to produce outputs in $[-1, 1]$: $y = \tanh(z), z \sim \mathcal{N}(\mu, \sigma^2)$. It is widely used in reinforcement learning over continuous bounded action spaces (Haarnoja et al., 2018) due to its support for the reparameterization trick.

Table 1 compares these bounded-interval supported distribution families across important properties for latent variable modeling. *Relative expressiveness* measures the variety of prototypical shapes a distribution can represent. All distributions except $\mathcal{CB}$ exhibit four prototypical shapes; $\mathcal{CB}$ is limited to two. Admittedly, none of these offer support for multimodal distributions over bounded intervals. Appendix H discusses other flexible alternatives (which have their own limitations), e.g., copulas (Tran et al., 2015) or normalising flows (Rezende & Mohamed, 2015). *Contains uniform* refers to the ability to represent the uniform distribution, critical for modeling complete uncertainty. All distributions except $\tanh_{\mathcal{N}}$ can express the uniform. *Closed-form CDF* indicates whether a closed-form CDF is available, which could be a desirable feature for use in copula-based modeling, simplifying the representation of complex dependencies (Tran et al., 2015). Only $\mathcal{CB}$ and KS provide such expressions. Similarly, *closed-form inverse CDF* indicates the availability of a closed-form inverse CDF, with only $\mathcal{CB}$ and KS satisfying this criterion. *Numerical issues* capture challenges in stable evaluation. For example, the $\mathcal{CB}$ log-pdf requires a Taylor expansion to handle singularities as $\lambda \to 0.5$. The $\tanh_{\mathcal{N}}$ distribution requires log-pdf clipping and parameter regularization to maintain stability, as appears in various implementations (Haarnoja et al., 2018). *Complex functions* highlight reliance on non-affine, non-logarithmic, or non-exponential operations. The $\tanh_{\mathcal{N}}$ log-pdf involves computing $\log\left(1 - \tanh^2(x)\right)$, which is numerically unstable (Björck et al., 2021). The Beta distribution relies on the Beta function and the regularized incomplete Beta function in its log-pdf and CDF, respectively, both requiring numerical approximations. In contrast, our novel parameterization of the KS distribution (Section 3) avoids complex functions in the log-pdf, CDF, and inverse CDF; note $a^{-1}$ is computed via $\exp(-\log a)$, avoiding division.

The KS differential entropy contains the digamma function $\phi^{(0)}$ and the trigamma function $\phi^{(1)}$, but because there exist rapidly converging polynomial expansions, they can be evaluated with very high accuracy in a fast, efficient, and numerically stable manner. *Parameterization* examines whether a distribution can be effectively expressed with unconstrained parameters. Both $\mathcal{CB}$ (via $\log \lambda \in \mathbb{R}$) and $\tanh_{\mathcal{N}}$ (via $(\mu, \log \sigma) \in \mathbb{R}^2$) support unconstrained parameterization. We introduce the first unconstrained parameterization for KS in Section 3, using $(\log a, \log b) \in \mathbb{R}^2$. The Beta distribution, due to its dependence on the Beta function, resists effective unconstrained parameterization. *Closed-form KL functions* refer to analytical KL divergence expressions. The $\mathcal{CB}$ and Beta distributions, as members of the exponential family, admit closed-form KL expressions with other exponential family members. The KS also has closed-form KL expressions with Beta family members, while $\tanh_{\mathcal{N}}$ is restricted to closed-form KL expressions within its own family. *Entropy* considers the availability of closed-form expressions for differential entropy. This property is present for all distributions except $\tanh_{\mathcal{N}}$.

**Latent variable modeling with stochastic variational inference (SVI).** The primary method for fitting large-scale latent variable models is SVI (Hoffman et al., 2013). Consider a model $p_{\boldsymbol{\theta}}(\boldsymbol{x}) = \int p_{\boldsymbol{\theta}}(\boldsymbol{x} \mid \boldsymbol{z})p(\boldsymbol{z})d\boldsymbol{z}$, where $\boldsymbol{x} \in \mathbb{R}^M$ is the observation, $\boldsymbol{z} \in \mathbb{R}^D$ is a vector-valued latent variable, $p_{\boldsymbol{\theta}}(\boldsymbol{x} \mid \boldsymbol{z})$ is the likelihood function with parameters $\boldsymbol{\theta}$, and $p(\boldsymbol{z})$ is the prior distribution. Except for a few special cases, maximum likelihood learning in such models is intractable because of the difficulty of the integrals involved. Variational inference (Jaakkola & Jordan, 2000) provides a tractable alternative by introducing a variational posterior distribution $q_{\boldsymbol{\phi}}(\boldsymbol{z})$ and maximizing a lower bound on the marginal log-likelihood called the ELBO:

$$\mathcal{L}(\boldsymbol{x}, \boldsymbol{\theta}, \boldsymbol{\phi}) = \mathbb{E}_{q_{\boldsymbol{\phi}}(\boldsymbol{z})}\left[\log p_{\boldsymbol{\theta}}(\boldsymbol{x} \mid \boldsymbol{z})\right] - D_{\mathrm{KL}}\left(q_{\boldsymbol{\phi}}(\boldsymbol{z}) \,\|\, p(\boldsymbol{z})\right) \leq \log p_{\boldsymbol{\theta}}(\boldsymbol{x}). \tag{1}$$

Training models with modern SVI (Kingma & Welling, 2014; Rezende et al., 2014) involves gradient-based optimization of this bound w.r.t. both the model parameters $\boldsymbol{\theta}$ and the variational parameters $\boldsymbol{\phi}$. The first term in (1) encourages the model to assign high likelihood to the data, but its exact evaluation and gradients are typically intractable and so the expectation is often approximated with samples from $q_{\boldsymbol{\phi}}(\boldsymbol{z})$. The KL divergence term incorporates prior information by penalizing deviations of the variational posterior from the prior $p(\boldsymbol{z})$. Closed-form expressions of $D_{\mathrm{KL}}\left(q_{\boldsymbol{\phi}}\left(\boldsymbol{z}\right) \,\|\, p\left(\boldsymbol{z}\right)\right)$ allow efficient encoding of prior information; otherwise, sample-based approximations are required. In the common setting of i.i.d. data with per-datapoint latent variables, amortized inference introduces a shared NN, parameterized by 'inference parameters' $\boldsymbol{\phi}$, to map observations to variational parameters, approximating their individual posteriors as $q_{\boldsymbol{\phi}}(\boldsymbol{z} \mid \boldsymbol{x})$. Modifying the ELBO by scaling the KL term with a parameter $\beta_{\mathrm{KL}} > 0$ is often necessary to balance the trade-off between data likelihood and prior regularization (Alemi et al., 2018). We denote the sample-based approximation of this modified ELBO as $\hat{\mathcal{L}}_{\beta_{\mathrm{KL}}}$.

**Gradient reparameterization: explicit and implicit.** A distribution $q_{\boldsymbol{\phi}}(\boldsymbol{z})$ is said to be *explicitly* reparameterizable, or amenable to the 'reparameterization trick', if it can be expressed as a deterministic, differentiable transformation $\boldsymbol{z} = g(\boldsymbol{\epsilon}, \boldsymbol{\phi})$ of a base distribution $\boldsymbol{\epsilon} \sim p(\boldsymbol{\epsilon})$. This base distribution is typically simple, such as Uniform or standard Normal, enabling fast sample generation by first sampling from the base and then applying $g$. This enables the use of backpropagation to estimate gradients of the form [cf. (1)]

$$\nabla_{\boldsymbol{\phi}}\mathbb{E}_{q_{\boldsymbol{\phi}}(\boldsymbol{z})}[f(\boldsymbol{z})] = \mathbb{E}_{p(\boldsymbol{\epsilon})}[\nabla_{\boldsymbol{\phi}}f(g(\boldsymbol{\epsilon}, \boldsymbol{\phi}))] = \mathbb{E}_{p(\boldsymbol{\epsilon})}[\nabla_{\boldsymbol{z}}f(\boldsymbol{z})|_{\boldsymbol{z}=g(\boldsymbol{\epsilon}, \boldsymbol{\phi})}\nabla_{\boldsymbol{\phi}}g(\boldsymbol{\epsilon}, \boldsymbol{\phi})], \tag{2}$$

an expectation with form encompassing the ELBO. Explicit reparameterization is compatible with distributions in the location-scale family (e.g., Gaussian, Laplace, Cauchy), distributions with tractable inverse CDFs (e.g., exponential, KS, $\mathcal{CB}$), or those expressible as deterministic transformations of such distributions (e.g., $\tanh_{\mathcal{N}}$). When explicit reparameterization is not available, implicit reparameterization (Figurnov et al., 2018) is commonly used for distributions with numerically tractable CDFs, such as truncated, mixture, Gamma, Beta, Dirichlet, or von Mises distributions. This method expresses the parameter gradient through the sample $\nabla_{\boldsymbol{\phi}}\boldsymbol{z}$ as a function only of the CDF gradients, not its inverse. Such CDF gradients are commonly found using numerical methods, e.g., forward mode auto-differentiation on truncated iterations to estimate the CDF, as in the Gamma and Beta distributions. Such numerical methods introduce approximation error, and thus potentially higher variance and numerical stability issues in the estimation of (2) (Mohamed et al., 2020), and tend to introduce significantly more complexity than explicit reparameterization which only require implementing $g(\epsilon, \phi)$.

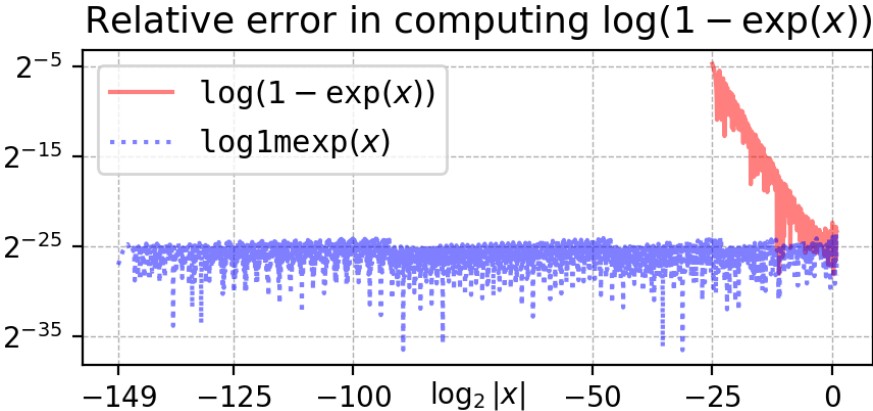

Figure 2: Naive computation of $\log{(1 - \exp{(x)})}$ (red) becomes unstable as $x \to 0$ due to catastrophic cancellation, while `log1mexp(x)` (blue) ensures accurate computation. Relative error is the absolute difference between single-precision and 1024-decimal-place evaluations, normalized by the 1024-decimal-place result.

## 3 Stabilizing the Kumaraswamy

The KS distribution's utility relies on stable computation of its log-pdf, CDF, inverse CDF, and their gradients. In the standard parameterization, these functions contain instabilities from hidden $\log(1 - \exp(x))$ terms. We address this by introducing an unconstrained logarithmic parameterization that isolates these unstable terms, enabling their straightforward replacement with the stable `log1mexp` function. Finally, we show why naive stabilization techniques, such as parameter clipping, fail in high-dimensional applications.

**Identifying the instability:** $\log{(1 - \exp{(x)})}$**.** Naive computation of $\log{(1 - \exp{(x)})}$ for $x < 0$ leads to significant numerical errors as $x$ approaches 0 (Figure 2, red). These errors grow so large that they can cause *numerical instability*, i.e., an irrecoverable error such as `-inf`. These errors result from *catastrophic cancellation*, which occurs when subtracting nearly equal numbers — here, $1 - \exp(x)$. As $x \to 0$, $\exp(x) \approx 1$, so `1 - exp(x)` results in the cancellation of leading significant bits, leaving only a few less significant, less accurate bits to represent the result. This causes large relative errors in `1 - exp(x)`, which are amplified when input to the logarithm as its magnitude grows sharply near zero. If the cancellation is complete, `1 - exp(x)` underflows to 0 and the logarithm returns `-inf`, as seen in Figure 2 (red) when $\log_2 |x| < -24$.

**Numerical building blocks for accurate** $\log{(1 - \exp{(x)})}$ **computation.** When $|x| \ll 1$, both $\log(1 + x)$ and $\exp(x) - 1$ can suffer from severe cancellation: the former between 1 and $x$, the latter between $\exp(x)$ and $-1$. In both cases, a simple solution for accurate computation is to use a few terms of the Taylor series, as

$$\texttt{log1p}(x) := \log(1 + x) = x - \frac{x^2}{2} + \frac{x^3}{3} - \ldots, \quad \text{for } |x| < 1,$$

$$\texttt{expm1}(x) := \exp(x) - 1 = x + \frac{x^2}{2!} + \frac{x^3}{3!} + \ldots, \quad \text{for } |x| < 1,$$

where $n!$ denotes the factorial. Standard single-precision `log1p` and `expm1` implementations typically use $\sim 4$ terms in their Taylor series, which we found sufficient for our needs. These functions form the basis for two common methods to compute $\log{(1 - \exp{(x)})}$: `log(-expm1(x))` and `log1p(-exp(x))`. Mächler (2012) showed that neither method alone provides sufficient accuracy across the domain, but each is accurate in complementary regions. To address this, Mächler (2012) introduced

$$\texttt{log1mexp}(x) := \begin{cases} \texttt{log(-expm1(x))} & -\log 2 \le x < 0 \\ \texttt{log1p(-exp(x))} & x < -\log 2, \end{cases} \tag{3}$$

which computes $\log{(1 - \exp{(x)})}$ accurately throughout single precision, shown in Figure 2 (blue).

**A stable Kumaraswamy.** The direct implementation of the KS's log-pdf, CDF, and inverse CDF — as found in all core auto-differentiation libraries — produces numerical instabilities. Here, we introduce a novel

parameterization in terms of unconstrained logarithmic parameter values, which isolates and makes explicit the unstable terms

$$w_{b^{-1}}(u) = \log(1 - u^{b^{-1}}) = \log(1 - \exp(b^{-1}\log u))$$
$$w_a(x) = \log(1 - x^a) \quad = \log(1 - \exp(a\log x)),$$

eliminates the need for positivity-enforcing link functions, and whose expressions involve only affine, exponential, and logarithmic transformations. This allows the log-pdf and its gradients to be expressed as

$$\log f(x) = \log a + \log b + (a-1)\log x + (b-1)w_a(x) \tag{4}$$
$$\nabla_{\log x}\log f(x) = (a-1) - (b-1)\cdot\exp(a\log x - w_a(x) + \log a) \tag{5}$$
$$\nabla_{\log a}\log f(x) = 1 + a\log x \cdot \{1 - (b-1)\cdot\exp(a\log x - w_a(x))\} \tag{6}$$
$$\nabla_{\log b}\log f(x) = 1 + b\cdot w_a(x). \tag{7}$$

Likewise for the CDF

$$F(x) = 1 - (1 - x^a)^b = 1 - \exp(b\cdot w_a(x)) \tag{8}$$
$$\nabla_x F(x) = \exp(\log a + \log b + (a-1)\cdot\log x + (b-1)\cdot w_a(x)) \tag{9}$$
$$\nabla_{\log a}F(x) = \exp(\log a + \log b + a\cdot\log x + (b-1)\cdot w_a(x))\cdot\log x \tag{10}$$
$$\nabla_{\log b}F(x) = \exp(\log b + b\cdot w_a(x))\cdot(-w_a(x)), \tag{11}$$

and the inverse CDF

$$F^{-1}(u) = (1 - u^{b^{-1}})^{a^{-1}} = \exp(a^{-1}w_{b^{-1}}(u)) \tag{12}$$
$$\nabla_{\log a}F^{-1}(u) = \exp(-\log a + a^{-1}w_{b^{-1}}(u))\cdot(-w_{b^{-1}}(u)) \tag{13}$$
$$\nabla_{\log b}F^{-1}(u) = \exp(-\log a - \log b + b^{-1}\log u + (a^{-1}-1)w_{b^{-1}}(u))\cdot\log u. \tag{14}$$

This parameterization's algebraic form allows direct replacement of the dominant unstable terms, substituting $w_{b^{-1}}(u)$ with $\texttt{log1mexp}\big(b^{-1}\log u\big)$ and $w_a(x)$ with $\texttt{log1mexp}(a\log x)$. Access to $\log a$ and $\log b$ avoids errors from unnecessary transitions in-and-out of log-space. We also avoid the error prone expressions produced in backpropogation's direct application of the chain rule, e.g.,

$$\nabla_{\log b}F^{-1} = \frac{1}{a}\cdot\exp\left(\frac{1}{a}\log\left(1\text{-}\exp\left(\frac{1}{b}\log u\right)\right)\right)\cdot\cdot\left(1\text{-}\exp\left(\frac{1}{b}\log u\right)\right)^{-1}\cdot\exp\left(\frac{1}{b}\log u\right)\cdot\log u\cdot\frac{-1}{b^2}\cdot b$$

and (14) are equivalent expressions for $\nabla_{\log b}F^{-1}$, but their computed values can differ greatly for extreme parameter values. Desirable KS distributions can obtain such problematic extreme parameter values, e.g., the KS distributions in Figure 3 have $b \approx 10^6$. See Section A for further discussion on how instability in the unmodified KS can worsen with increasing evidence.

Both PyTorch and TensorFlow implement the KS distribution similarly and fail to address the $\texttt{log1mexp}$ instabilities we highlight. Here, we use PyTorch as a representative example, though our results and methods apply equally to TensorFlow. Figure 3 compares the PDF, inverse CDF, and histograms of reparameterized samples for KS distributions which are typical to real-world modeling scenarios. The PyTorch implementation (top row) shows jaggedness in both the PDF and inverse CDF, caused by catastrophic cancellation in the unstable terms $w_a(x)$ and $w_{b^{-1}}(u)$. Additionally, the PyTorch inverse CDF underflows beyond $u \approx 1 - 39.3$: here, $w_{b^{-1}}(u) = -\infty$, and $F^{-1}(u) = \exp(a^{-1}\cdot-\infty) = 0$. This underflow results in a point mass at $x = 0$ (a point outside of the KS support) with probability $\approx 39.3$ in each of the reparameterized sampling distributions, and produces infinite gradients via $\nabla_{\log a}F^{-1} = \infty$ [cf. (13)]. This infinite gradient triggers a cascade: infinite parameter values after the optimizer step and $\texttt{NaN}$ activations in the next forward pass, which is what users ultimately observe when training fails.

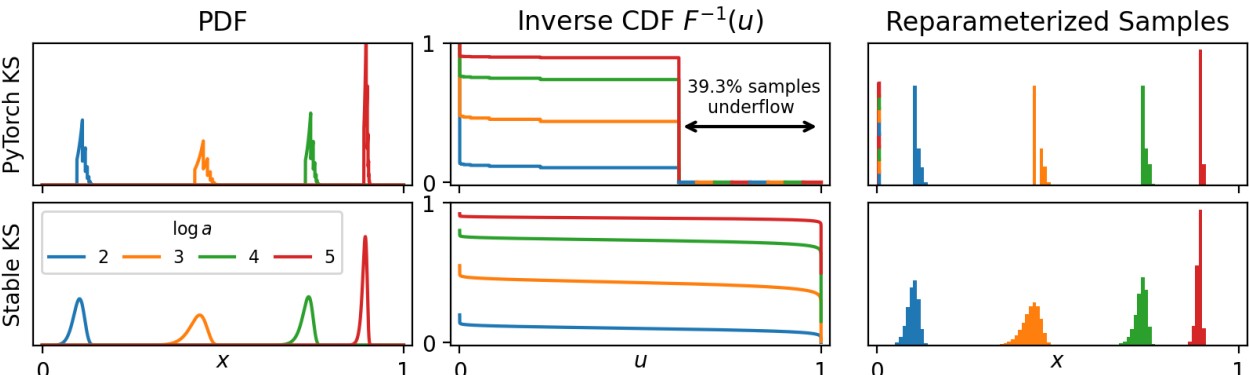

Figure 3: Stabilizing $\log(1 - \exp(x))$ terms eliminates numerical instabilities in the KS log-pdf and inverse CDF. We compare the unstable PyTorch KS implementation (top row) and our stable KS (bottom row) for realistic KS distributions ($\log_2 b = 24$, varying $a$). Catastrophic cancellation in the $\log(1 - \exp(x))$ terms in the PyTorch KS causes jagged curves and inverse CDF underflow beyond $u \approx 1 - 39.3$, resulting in a point mass of $\approx 39.3$ at $x = 0$ in the sampling distribution. Our stable KS removes the instability by using `log1mexp`.

### 3.1 The inadequacy of parameter clipping in large-scale settings

Numerical instability in the KS is inherently stochastic, and in high-dimensional settings, the compounded probability of failure across multiple variables makes program failure almost certain. As the program goes unstable if any single KS goes unstable, the overall instability probability can be modeled as the probability of at least one failure in $D$ independent Bernoulli trials: $1 - p(\text{KS stable})^D$, for $D$ KS latent variables. In practical large-scale settings, e.g., recommendation systems with $10^7$ users and $D = 10^7$ recommendation items, the probability of instability approaches 1 across all reasonable parameter clipping values, rendering clipping an ineffective stabilization strategy.

*Quantitative illustration.* Consider the stochastic instability arising from the term $\log(1 - \exp(b^{-1} \log u))$, where catastrophic cancellation occurs if $1 - \exp(b^{-1} \log u)$ becomes too small. To avoid logarithmic domain errors in single precision, we enforce $-b^{-1} \log u > 2^{-24}$ (Figure 2). We aim to select a $b_{\max}$ to satisfy this constraint: a larger $b_{\max}$ expands the variational family allowing improved posterior approximation, but worsens stability. Consider the moderate entropy KS distributions in Figure 3 which use $b = 2^{24}$. Using $b_{\max} = 2^{24}$, only $u < 0.6321$ satisfies the stability condition, i.e., $p(\text{KS stable}) \approx 0.6321$ per sample. With $D = 10^7$, the overall probability of instability becomes $1 - 0.6321^{10^7} \approx 1$. Now consider aggressively restricting $b_{\max} = 2^4$, as done in (Nalisnick et al., 2016). Now $u < 0.9999$ satisfies the stability condition. Even then, introducing $D = 10^7$ variables, we still have the overall probability of instability is $1 - 0.9999^{10^7} \approx 1$. Thus, even extreme clipping fails to stabilize KS distributions at scale. Further, this analysis considers only a single posterior sample. In practice, training with SVI requires $T \sim 10^3$ optimization steps, each requiring posterior samples for gradient estimation. This compounds the instability probability to $1 - p(\text{KS stable})^{TD}$, making clipping ineffective in realistic large-scale scenarios.

## 4 Experiments

Using the well established VAE framework on MNIST and CIFAR-10 datasets, we show that the stabilized KS enables reliable training as both a variational posterior [Eqns. (12)–(14)] and likelihood function [Eqns. (4)–(7)]. We then introduce two new deep variational architectures that leverage bounded interval-supported latent variables: the Variational Bandit Encoder (VBE) for addressing exploration-exploitation trade-offs in contextual multi-armed bandits (Section 4.2), and the Variational Edge Encoder (VEE) for facilitating uncertainty quantification in link-prediction with graph neural networks (Section 4.3). Across the experimental domains, our stable KS is performant and often easier to use than alternative variational distributions supported on bounded intervals. For instance, $\tanh_{\mathcal{N}}$ models require log-pdf clipping for stability,

Table 2: VAE on MNIST and CIFAR-10.

| Prior | $q_\phi(z|x)$ | $p_\theta(x|z)$ | MNIST | | CIFAR-10 | |
|---|---|---|---|---|---|---|
| | | | LL | Acc. | LL | Acc. |
| $\mathcal{N}_{(0,1)}$ | $\mathcal{N}$ | $\mathcal{CB}$ | **1860** | 97.3 | 1191 | 37.9 |
| $U_{(0,1)}$ | KS | $\mathcal{CB}$ | 1855 | 97.4 | **1194** | **41.5** |
| $U_{(0,1)}$ | Beta | $\mathcal{CB}$ | 1856 | **97.5** | 1189 | 40.3 |
| $\mathcal{N}_{(0,1)}$ | $\mathcal{N}$ | Beta | 4209 | **92.1** | **3648** | 48.5 |
| $U_{(0,1)}$ | KS | Beta | 4160 | 91.3 | 3579 | **50.1** |
| $U_{(0,1)}$ | Beta | Beta | **4198** | 90.1 | N/A | N/A |
| $\mathcal{N}_{(0,1)}$ | $\mathcal{N}$ | KS | 3393 | 96.4 | 1875 | 47.1 |
| $U_{(0,1)}$ | KS | KS | 3401 | 96.8 | **1939** | **48.8** |
| $U_{(0,1)}$ | Beta | KS | **3405** | **97.1** | N/A | N/A |

Table 3: MNIST test digit VAE reconstructions.

while Beta models show significant performance variability based on the chosen positivity-enforcing link function and often fail to converge, e.g., on CIFAR-10 in Section 4.1. Finally, our new variational models are fast: the VBEs in Section 4.2 are $8 - 22\times$ faster than the state-of-the-art baseline.

**Remark.** *Across all three experimental settings, models using the unstable KS produce* `NaN` *errors in training and are therefore excluded.* Prior work using the KS in low-dimensional latent variable models (Nalisnick et al., 2016; Nalisnick & Smyth, 2017; Stirn et al., 2019) similarly find `NaN` errors, and rely on parameter clipping to avoid instability. See Section 3.1 for why this is approach does not work in large-scale settings. Our stabilization approach directly resolves these numerical issues, enabling stable training at scale.

## 4.1 Image variational auto-encoders

The VAE (Kingma & Welling, 2014) is a generative latent variable model trained using amortized variational inference. Both the variational posterior $q_\phi(z|x)$ and the conditional likelihood $p_\theta(z \mid x)$ are parameterized using NNs, known as the encoder $e_\phi(x) : \mathbb{R}^M \mapsto \mathbb{R}^D$ and decoder $d_\theta(z) : \mathbb{R}^D \mapsto \mathbb{R}^M$, respectively. VAEs typically use the standard Normal distribution as the prior and a factorized Normal as the variational posterior. The use of alternative variational distributions allows incorporating different prior assumptions about the latent factors of the data, such as bounded support or periodicity (Figurnov et al., 2018).

**Experimental setup and metrics.** Inspired by (Loaiza-Ganem & Cunningham, 2019), we train VAEs with fully factorized priors and variational posteriors on MNIST and CIFAR-10 without pixel binarization, using an unmodified ELBO ($\beta_{\mathrm{KL}} = 1$). We adopt the most effective likelihoods from their work (Beta and $\mathcal{CB}$), identical latent dimension $D$ (MNIST: $D = 20$, CIFAR-10: $D = 50$), and the same standard NN architectures, which are detailed in Appendix B, along with the training hyperparameters. For each variational posterior factor, we choose the canonical prior: $\mathcal{N}_{(0,1)}$ for $\mathcal{N}$, and $U_{(0,1)}$ for KS and Beta. We evaluate the models with an importance weighted estimator (Burda et al., 2016) of the marginal Log Likelihood (LL) using 200 samples. To assess usefulness of the learned latent representations, we encode test data $x_n$, compute the mean of $q_\phi(z_n \mid x_n)$, and classify the test labels using a 15-nearest neighbor classifier; the classifier accuracy (%) is denoted "Acc." For subjective evaluation, we display the mean decoded likelihood of a single sample from the encoded posterior of random test digits in Figure 3.

**Discussion of results.** Rather than introducing a more performant VAE architecture, the sole purpose of this experiment is to provide evidence toward the stabilization of the KS. Notably, stable KS VAEs maintain numerical stability while all VAEs with the unstable KS produce unstable training. VAEs with Beta-distributed variational posteriors often do not converge; indeed, (Figurnov et al., 2018) reported strong performance on binarized MNIST using a softplus link function, but did not present results on CIFAR-10,

nor could we find other works that did. We suspect this is due to similar instability issues, with the higher gradient variance of the Beta's implicit reparameterization a likely explanation. In an attempt to overcome this instability in Beta VAEs we report the best metrics across softplus or exp link functions in Table 2. When neither converges, we report N/A. The results in Table 2 show that across datasets, VAEs with KS-distributed variational posteriors consistently produce useful latent spaces, evidenced by strong latent nearest neighbor classifier accuracy, and yield reconstructions with competitive LLs and visual quality.

When paired with any variational posterior, a KS likelihood yields stronger MNIST reconstructions than Beta likelihoods: compare rows $*$-Beta to $*$-KS in Table 3. As in (Loaiza-Ganem & Cunningham, 2019), we find $\mathcal{CB}$ likelihoods produce the most subjectively performant VAEs on MNIST, unsurprising as $\mathcal{CB}$ was introduced specifically for the approximately binary MNIST pixel data.

## 4.2 Contextual Bernoulli multi-armed bandits

The contextual Bernoulli multi-armed bandit (MAB) problem involves a decision maker who, at each time step $t = 1, \ldots, T$, selects one arm from a finite set of $K$ options. Each arm has an associated context $\boldsymbol{x}_k \in \mathbb{R}^d$, and pulling an arm yields a binary reward $r_k \sim \text{Bernoulli}(\mu_k)$, where $\mu_k \in [0, 1]$ is the unknown mean reward. MABs originate by analogy to casino slot machines, where each machine (arm) has a different payout rate, and the challenge lies in deciding which arms to pull in order to maximize total winnings while learning about their payout rates, a situation called the exploration-exploitation dilemma. MABs have found applications in modern recommendation systems (Li et al., 2010), clinical trials design (Villar et al., 2015), and mobile health (Tewari & Murphy, 2017). Thompson Sampling (TS) is a simple, empirically effective (Chapelle & Li, 2011), and scalable (Jun et al., 2017) arm selection heuristic. It selects the arm corresponding to the highest value drawn from the posterior distributions over the latent $z_k$'s. This approach naturally balances exploration and exploitation: the uncertainty in the posteriors promote exploration, while concentration of probability mass on large mean rewards drive exploitation. See Section 5 for further discussion on TS-based Bernoulli MAB approaches.

**Variational Bandit Encoder (VBE).** Consider $K$ arms, each with context $\boldsymbol{x}_k \in \mathbb{R}^d$ and an associated latent variable $z_k \in [0, 1]$ representing its mean reward. Define $\boldsymbol{X} = [\boldsymbol{x}_1, \ldots, \boldsymbol{x}_K]$. Let $\mathcal{T}_k$ be the set of time indices at which arm $k$ is chosen, resulting in binary rewards $\{r_{k,t}\}_{t \in \mathcal{T}_k}$. The joint distribution factorizes as

$$p_{\boldsymbol{\theta}}(\boldsymbol{z}, \boldsymbol{r} \mid \boldsymbol{X}) = \prod_{k=1}^{K} \left[ p_{\boldsymbol{\theta}}(z_k \mid \boldsymbol{x}_k) \prod_{t \in \mathcal{T}_k} \text{Bernoulli}(r_{k,t} \mid z_k) \right], \tag{15}$$

where $p_{\boldsymbol{\theta}}(z_k \mid \boldsymbol{x}_k)$ is a conditional prior on the latent mean reward $z_k$; see Appendix F for strategies to incorporate prior knowledge into this distribution. In the following, we take $p_{\boldsymbol{\theta}}(z_k \mid \boldsymbol{x}_k)$ to be uniform, and so $\boldsymbol{\theta} = \emptyset$. Each reward $r_{k,t} \in \{0, 1\}$ is drawn from Bernoulli($\mu_k$), so if an arm is never pulled ($\mathcal{T}_k = \emptyset$), it simply yields no observed rewards.

With regards to variational approximations and learning, VBE's posit a fully factorized KS variational posterior $\prod_k q_{\boldsymbol{\phi}}(z_k \mid \boldsymbol{x}_k)$. Similar to VAEs, we employ amortized inference using a shared NN encoder $e_{\boldsymbol{\phi}}(\boldsymbol{x}_k)$, which defines a reparameterizable variational distribution $q_{\boldsymbol{\phi}}(z_k \mid \boldsymbol{x}_k)$. However, unlike VAEs, VBEs omit the decoder; samples $\tilde{z}_k \sim q_{\boldsymbol{\phi}}(z_k \mid \boldsymbol{x}_k)$ directly parameterize the likelihood function. The arm selection at step $t$ follows TS: $a = \text{argmax}_k\{\tilde{z}_k\}$. We then draw reward $r \sim \text{Bernoulli}(\mu_a)$ and record it in the replay buffer $\mathcal{D} \leftarrow \mathcal{D} \cup \{(\boldsymbol{x}_a, a, r)\}$. We construct a sample approximation of the modified ELBO over the subset of arms $\mathcal{K}_t \subset \{1, \ldots, K\}$ that have been pulled by time $t$ as

$$\hat{\mathcal{L}}_{\beta_{\text{KL}}, t}(\mathcal{D}, \boldsymbol{\phi}) = \sum_{(\boldsymbol{x}_a, a, r) \in \mathcal{D}} \log p(r \mid \tilde{z}_a) + \beta_{\text{KL}} \sum_{k \in \mathcal{K}_t} \mathcal{H}[q_{\boldsymbol{\phi}}(z_k \mid \boldsymbol{x}_k)], \tag{16}$$

see Appendix E for the derivation. The second term promotes exploration by penalizing overconfidence with the exploration effect proportional to $\beta_{\text{KL}}$. We maximize (16) w.r.t. $\boldsymbol{\phi}$ via gradient ascent, enabled by the reparameterizable KS. VBE execution is summarized in Algorithm 1.

**VBE advantages.** VBEs provide four primary advantages over alternative TS-based Bernoulli MAB approaches, discussed in Section 5

**Algorithm 1** Variational Bandit Encoder

**Require:** $\{\boldsymbol{x}_k\}_{k=1}^K$, $\{\mu_k\}_{k=1}^K$, $\eta$, $\beta_{\mathrm{KL}}$

1: Variation posterior $q \leftarrow$ KS
2: Replay buffer $\mathcal{D} \leftarrow \emptyset$
3: **for** $t = 1 \ldots T$ **do**
4:     Encode: $(a_k, b_k) = e_{\boldsymbol{\phi}}(\boldsymbol{x}_k)$
5:     Sample: $\tilde{z}_k \sim q(z_k; a_k, b_k)$
6:     TS: $a = \operatorname{argmax}_k \{\tilde{z}_k\}$
7:     Reward: $r \sim \text{Bernoulli}(\mu_a)$
8:     $\mathcal{D} \leftarrow \mathcal{D} \cup \{(\boldsymbol{x}_a, a, r)\}$
9:     Construct $\hat{\mathcal{L}}_{\beta_{\mathrm{KL}}}$ as in (16)
10:    $\boldsymbol{\phi} \leftarrow \boldsymbol{\phi} + \eta \nabla_{\boldsymbol{\phi}} \hat{\mathcal{L}}_{\beta_{\mathrm{KL}}}$
11: **end for**

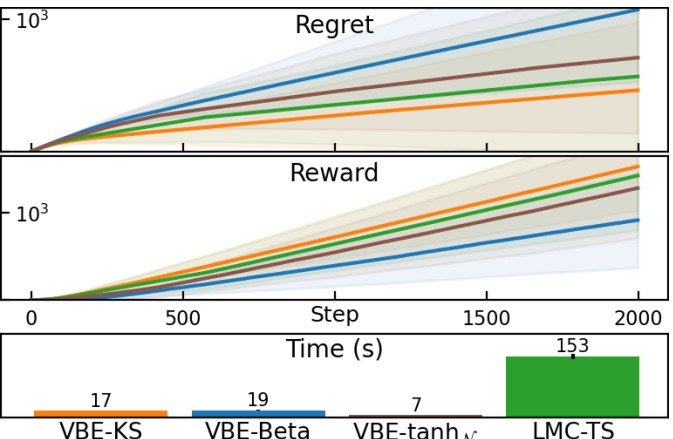

Figure 4: Synthetic bandit performance over 5 runs. VBE-KS best handles exploration-exploitation trade-offs.

- *Scalability and Compatibility.* VBE training consists of a forward pass through a NN, sampling an explicitly reparameterized distribution, and a backward pass for gradient-based updates. This process is scalable and fully compatible with existing gradient-based infrastructure.

- *Prior Knowledge Incorporation.* When prior knowledge exists on an arm $k$ we can encode it as $p_{\boldsymbol{\theta}}(z_k \mid \boldsymbol{x}_k) = \text{Beta}(a_k, b_k)$, replacing $\mathcal{H}[q_{\boldsymbol{\phi}}(z_k \mid \boldsymbol{x}_k)]$ with $-D_{\mathrm{KL}}\left(q_{\boldsymbol{\phi}}(z_k \mid \boldsymbol{x}_k) \,\|\, p_{\boldsymbol{\theta}}(z_k \mid \boldsymbol{x}_k)\right)$; see Appendix F.

- *Interpretability and Independence.* Encoding $\boldsymbol{x}_k$ with $e_{\boldsymbol{\phi}}$ produces KS distribution parameters, fully encapsulating the model's beliefs about $\mu_k$. This is independent of other arms and past data.

- *Simplicity.* VBEs lack numerous hyperparameters and complex architectural components.

Alternative methods lack some or all of these properties because they do not directly model the mean rewards nor differentiate through reparameterized samples from a variational mean reward posterior; instead, they tend to model parameters of a context-to-mean reward function.

**Experimental setup.** We construct synthetic data with $K = 10^4$ arms by first sampling a weight vector $\mathbf{w}$ and features $\{\boldsymbol{x}_k\}_{k=1}^K$ from $\mathcal{N}(\mathbf{0}, \mathbf{I}_5)$. We then compute $\{\mathbf{w}^\top \boldsymbol{x}_k\}_{k=1}^K$ and apply min-max normalization to produce probabilities (referred to as "Original probabilities" in Figure 5). To introduce non-linearity, we raise these probabilities to the power 5 (shown as "Power (5) transformed probabilities" in Figure 5). Exponentiating the probabilities not only makes the mapping from features to mean rewards more challenging to learn, but it also significantly reduces the number of arms with high probabilities, forcing the agent to explore more. For instance, when raising the probabilities to the power of 5, the number of arms with large probabilities drops from 167 to just 7. We consider $T = 2 \cdot 10^3$ steps.

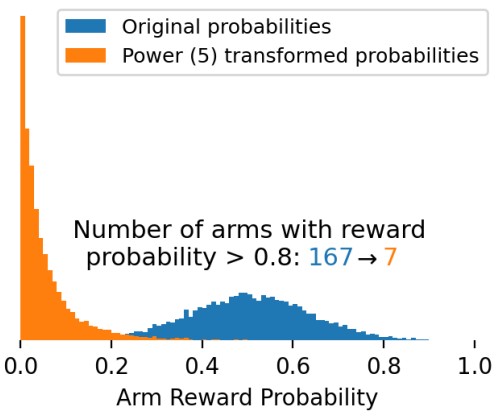

Figure 5: High arm reward probabilities are reduced via a power 5 exponentiation, encouraging exploration.

We evaluate VBEs with either a KS (VBE-KS), Beta (VBE-Beta), or $\tanh_{\mathcal{N}}$ (VBE-$\tanh_{\mathcal{N}}$) all using $\beta_{\mathrm{KL}} = |\mathcal{K}_t|^{-1}$, which makes the second term in (16) a mean. Gaussian variational posteriors are not used as their support $\mathbb{R}^K$ does not match the posterior support $[0,1]^K$. VBE-$\tanh_{\mathcal{N}}$'s performance is sensitive to the number of samples used in its entropy estimate: we found degraded performance beyond 10 samples. The learning rate is set to $\eta = 10^{-2}$. As a baseline, we use LMC-TS, which employs Langevin Monte

Carlo (LMC) to sample posterior parameters of a NN, known for state-of-the-art performance across various tasks (Xu et al., 2022). All models use an MLP with 3 hidden layers of width 32. LMC-TS hyperparameters (inverse temperature, LMC steps, weight decay) are set or tuned based on the authors' code. We repeat experiments 5 times on an Apple M2 CPU and report the mean and standard deviation across these runs in Figure 4.

**Metrics and evaluation.** The optimal policy always selects the arm with the highest mean reward $r^*$. Our objective is to minimize regret, defined as the cumulative difference between the expected reward of the chosen action and the optimal action (accessible in the synthetic setting), i.e., $\sum_{t=1}^{T}(r^* - r_{a_t})$. VBE-KS achieves lower regret and higher cumulative reward than all baselines. VBE-Beta performs significantly worse than VBE-KS and VBE-tanh$_{\mathcal{N}}$, highlighting the importance of explicit reparameterization. LMC-TS is performant but 8–22$\times$ slower than VBEs: VBEs avoid the computational overhead of LMC.

## 4.3 Variational link prediction with Graph Neural Networks

Graph Neural Networks (GNNs) have become a powerful tool for learning from graph-structured data, with applications in critical areas like drug discovery (Zhang et al., 2022) and finance (Wang et al., 2022). A key task is link prediction, where the goal is to infer unobserved edges between nodes. However, real-world deployment of graph learning models is often hindered by a lack of reliable uncertainty estimates and limited capacity to incorporate prior knowledge (Wasserman & Mateos, 2024). To address these challenges, we propose a variational approach where the GNN encodes a fully factorized (across the edge set) KS variational posterior to model the unobserved probabilities of each network link's existence, enabling uncertainty quantification and prior knowledge integration with minimal computational overhead.

In a typical link prediction setup, the GNN has access to the features $\boldsymbol{X} \in \mathbb{R}^{N \times d}$ of all $N$ nodes, but only a subset of positive edges in the training $\mathcal{D}_{tr}$ and validation $\mathcal{D}_{val}$ sets. Edge $(u, v)$ takes value $l_{u,v} = 1$ when present, or $l_{u,v} = 0$ when absent. The GNN generates edge embeddings through message passing and neighborhood aggregation, outputting probabilities $z_{u,v} \in (0, 1)$ that parameterize a Bernoulli likelihood. The seminal work of (Kipf & Welling, 2016) proposed Variational Graph Auto-encoders (VGAEs), which posits a Gaussian variational posterior over the final *node* embeddings. When used for link prediction it samples final node embeddings from the variational posterior and decodes them to produce edge probabilities. In contrast, our approach directly models the probability of an edge using the KS. An advantage of directly modeling edge probabilities is interpretability; deep nodal embeddings are often difficult to interpret, and priors are typically selected for computational tractability rather than their ability to incorporate meaningful prior information. However, the probability of an edge $(u, v)$ existing between two nodes is an interpretable quantity that can often be informed by domain expertise , and incorporated into the conditional edge prior $p_{\boldsymbol{\theta}}(z_{u,v} \mid \boldsymbol{x}, \mathcal{D}_{tr})$. For example, in gene regulatory networks, epidemiological networks, and social networks experts often have prior knowledge about the likelihood of specific interactions, transmissions, or friendships, respectively. We believe the limited exploration of variational modeling for edge probabilities is due to the previous lack of an expressive, stable, explicitly reparameterizable bounded-interval distributions.

**Variational Edge Encoder (VEE).** Let $\mathcal{E}_{tr}$ be the edges included in the training set, which consist of positive edges $\mathcal{D}_{tr}$ and a set of sampled negative edges $\mathcal{D}_{tr}^-$. The joint distribution factorizes as

$$p_{\boldsymbol{\theta}}(\boldsymbol{z}, \mathcal{E}_{tr} \mid \boldsymbol{x}) = \prod_{(u,v) \in \mathcal{E}_{tr}} p_{\boldsymbol{\theta}}(z_{u,v} \mid \boldsymbol{x}, \mathcal{E}_{tr}) \, \text{Bernoulli}(l_{u,v} \mid z_{u,v}). \tag{17}$$

Here, we take $p_{\boldsymbol{\theta}}(z_{u,v} \mid \boldsymbol{x}, \mathcal{E}_{tr})$ to be uniform, and so $\boldsymbol{\theta} = \emptyset$. *Variational approximations and learning.* We propose a fully factorized KS variational posterior $\prod_{(u,v) \in \mathcal{E}_{tr}} q_{\boldsymbol{\phi}}(z_{u,v} \mid \boldsymbol{X}, \mathcal{E}_{tr})$. The GNN encoder $e_{\boldsymbol{\phi}}$ parameterizes a KS distribution for each possible edge $(u, v) \in \mathcal{E}_{tr}$. The remaining structure is highly similar to VBEs: a single sample $\tilde{z}_{u,v} \sim q_{\boldsymbol{\phi}}(z_{u,v} \mid \boldsymbol{X}, \mathcal{E}_{tr})$ directly parameterizes the Bernoulli likelihood, and we maximize a sample approximation of the modified ELBO

$$\hat{\mathcal{L}}_{\text{KL}}((\boldsymbol{X}, \mathcal{E}_{tr}), \boldsymbol{\phi}) = \sum_{(u,v) \in \mathcal{E}_{tr}} \log p(l_{u,v} \mid \tilde{z}_{u,v}) + \beta_{\text{KL}} \sum_{(u,v) \in \mathcal{E}_{tr}} \mathcal{H}[q_{\boldsymbol{\phi}}(z_{u,v} \mid \boldsymbol{X}, \mathcal{E}_{tr})]. \tag{18}$$

From their similarity with VBEs, VEEs inherit the same advantages outlined in Section 4.2.

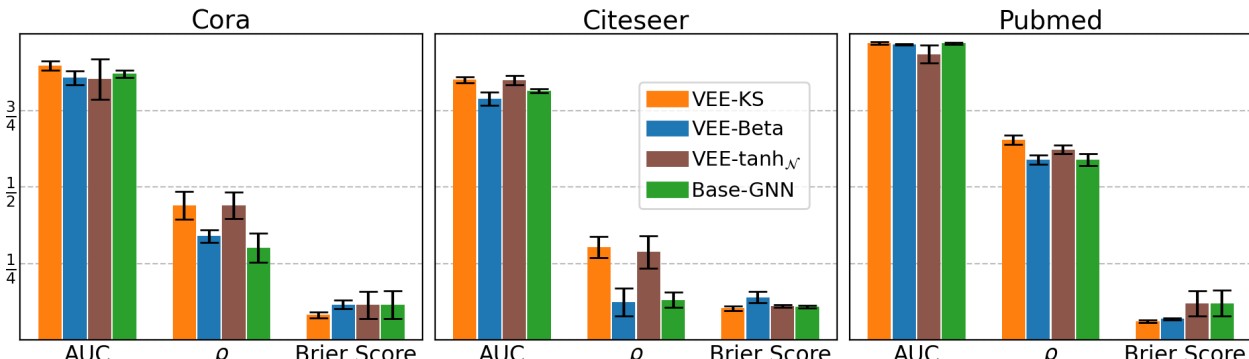

Figure 6: VEE-KS produces high predictive accuracy ($\uparrow$ AUC), informative uncertainty estimates ($\uparrow \rho$), and improved calibration ($\downarrow$ Brier Score) across graph datasets.

**Models, metrics, and datasets.** All models use a 2-layer GNN with Graph Convolutional Network (GCN) layers and a hidden/output nodal dimension of 32. In Base-GNN, an MLP decodes the final nodal embeddings into link probabilities. In VEE-KS/Beta/tanh$_\mathcal{N}$ an MLP parameterizes the KS/Beta/tanh$_\mathcal{N}$ variational distributions; all take $\beta_{\mathrm{KL}} = .05|\mathcal{D}_{tr}|^{-1}$. We use 10 samples in tanh$_\mathcal{N}$'s entropy estimate; more did not produce significant performance differences. We train for 300 epochs, with a learning rate of .01, averaging results over 5 runs with different seeds. The posterior predictive distribution over binary links $p(\boldsymbol{A} \mid \boldsymbol{X}, \mathcal{E}_{tr}) = \int p(\boldsymbol{A}|\boldsymbol{z})q_{\boldsymbol{\phi}}(\boldsymbol{z} \mid \boldsymbol{X}, \mathcal{E}_{tr})d\boldsymbol{z}$ is estimated by using a single sample from each KS/Beta distribution, parameterizing each edge Bernoulli distribution with such samples, followed by sampling binary edges. For Base-GNN we directly sample binary edges from the likelihood. Using 30 posterior predictive samples, we compute the edge-wise posterior predictive mean (pred. mean) and standard deviation (pred. stdv.). We measure predictive accuracy using the area under the ROC curve (AUC) (higher is better), treating pred. mean as the predictor. To assess uncertainty estimation quality, we report the Pearson correlation $\rho$ (higher is better) between the predictive uncertainty (pred. stdv.) and error (the $\ell_1$ difference between pred. mean and the true label). A strong positive correlation indicates that higher uncertainty corresponds to greater predictive error, signifying informative uncertainty estimates. Finally, we evaluate model calibration using the Brier Score (lower is better), defined as the squared difference between the predicted probability and the true label. Figure 6 presents results across three standard citation networks: Cora, Citeseer, and Pubmed.

**Discussion of results.** On all datasets and all metrics, VEE-KS outperforms or is comparable to the most performant baselines, often providing higher predictive accuracy, better uncertainty estimation, and improved calibration. Similar to Section 4.2, we find Beta distributed variational posteriors perform significantly worse than those using KS or tanh$_\mathcal{N}$, further underlining the importance of explicit reparameterization. Moreover, models using explicitly reparameterizable latents are faster: on the largest dataset (Pubmed), the average time (ms) per epoch for VEE-KS, VEE-tanh$_\mathcal{N}$, and VEE-Beta was $381 \pm 61$, $301 \pm 26$, and $447 \pm 86$ respectively, on an Apple M2 CPU.

## 5 Related Work

**VBEs in context: TS-based Bernoulli MAB approaches.** Existing TS-based approaches for Bernoulli MABs assume a prior over model parameters $p(\boldsymbol{\phi})$, which map contexts to rewards through $e_{\boldsymbol{\phi}}$. At each round, parameters are sampled from the posterior, $\tilde{\boldsymbol{\phi}}_t \sim p(\boldsymbol{\phi} \mid \mathcal{D})$, and used to compute mean reward posterior samples $\{e_{\tilde{\boldsymbol{\phi}}_t}(\boldsymbol{x}_k)\}_{k=1}^K$. However, the Bernoulli likelihood often leads to intractable posteriors, making parameter sampling difficult. Common methods use either variational approximations (Chapelle & Li, 2011; Urteaga & Wiggins, 2018; Clavier et al., 2024), primarily Laplace, or MCMC approaches like Gibbs sampling (Dumitrascu et al., 2018) or LMC (Xu et al., 2022). These approaches face several limitations. First, incorporating prior knowledge is challenging since the relationship between a parameter's value and its effect on rewards is often unclear, except in the simplest models. Second, scalability is a concern: Laplace

approximations become inefficient with large context dimensions or model sizes, while MCMC-based methods are compute and memory intensive, requiring long burn-in periods (typically $10^2$ iterations) and large machine memory to store the buffer $\mathcal{D}$. Third, interpreting model beliefs over mean rewards requires drawing numerous posterior samples, adding further computational cost. Finally, these methods often introduce significant complexity through intricate algorithms, architectures, optimization steps, and hyperparameters, particularly MCMC parameters (e.g., burn-in iterations, chain length, LMC inverse temperature/weight decay and their respective schedules). By directly modeling mean rewards with a KS, instead of the parameters $\phi$, VBEs offer a simple, scalable, and interpretable approach to Bernoulli MABs.

**Kumaraswamy as a Beta surrogate.** A simple approach to overcome the Beta distribution's lack of explicit reparameterization is to use the KS as a surrogate. This surrogate approach is feasible due to their significant similarities when defined by the same two parameters and the availability of a high-fidelity closed-form approximation of the KL divergence between Beta and KS distributions. (Nalisnick et al., 2016; Nalisnick & Smyth, 2017) use KSs as surrogates for Betas in the Dirichlet Process stick-breaking construction to allow for stochastic latent dimensionality in a VAE. However, both require parameter clipping for numerical stability. In their published code (Nalisnick et al., 2016) constrains KS parameters $\log a, \log b \in [-2.3, 2.9]$, significantly limiting the expressiveness of latent KS distributions. Also, (Nalisnick & Smyth, 2017) comments under a *Computational Issues* section that 'If NaNs are encountered...clipping the parameters of the variational Kumaraswamys usually solve the problem.' (Stirn et al., 2019) improved upon (Nalisnick et al., 2016) by resolving the order-dependence issue in approximating a Beta with a KS. Similarly, (Singh et al., 2017) followed a comparable process using an Indian Buffet Process. Both works maintained numerical stability by restricting the uniform base distribution's support from the unit interval to a narrower interval, before passing the samples through the inverse CDF producing a distortion of the reparameterized sampling distribution. This work eliminates the need for such distortions, enabling more accurate Beta approximations and simplifying the use of the KS distribution by ensuring numerical stability without additional interventions.

## 6  Conclusion, Limitations, and Future Work

We identified and resolved key numerical instabilities in the KS distribution, a uniquely attractive option in variational models for bounded latent variables. Our work demonstrates that the stabilized KS can tackle a wide range of large-scale machine learning challenges by powering simple deep variational models. We introduce the Variational Bandit Encoder, which enhances exploration-exploitation trade-offs in contextual Bernoulli MABs, and the Variational Edge Encoder, which improves uncertainty quantification in link prediction using GNNs. Our empirical results show these models are both performant and fast, achieving their best performance with the KS while avoiding the instability and complexity seen in alternatives like the Beta or $\tanh_{\mathcal{N}}$ distributions. These models open avenues for addressing other large-scale challenges, including in recommendation systems, reinforcement learning with continuous bounded action spaces, network analysis, and uncertainty quantification in deep learning, such as modeling per-parameter dropout probabilities using a Concrete distribution (Gal et al., 2017).

KS generalizations (Usman & ul Haq, 2020) inherit $\log(1 - \exp(x))$ instabilities, which future work can resolve by building on our stabilization technique. A limitation of the current models is their inability to capture multimodal posteriors. Future work could explore KS mixtures or hierarchical latent spaces to bridge this gap. Further, optimizing the $\beta_{\mathrm{KL}}$ parameter with techniques like warm-up schedules could yield further performance gains (Alemi et al., 2018). Applications of our stable KS distribution to non-parametric models like the Dirichlet Processes follows directly from prior work (Nalisnick & Smyth, 2017; Stirn et al., 2019). Lastly, a theoretical analysis of the VBE, particularly in proving regret bounds and asymptotic results, could extend its applicability to critical areas like clinical trials, where robust decision-making under uncertainty is essential.

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

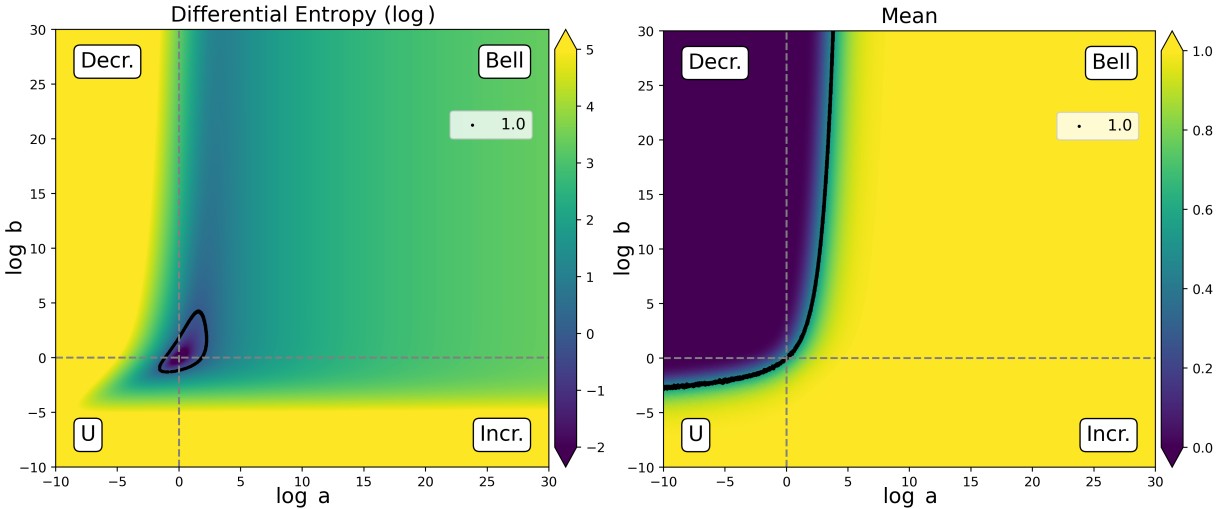

Figure 7: Differential entropy and mean of Kumaraswamy distributions across a wide range of parameter values. Low-entropy distributions are concentrated near the origin, where $\log a = \log b = 1$ corresponds to the uniform distribution. Distributions with a log differential entropy of 1 and mean of 0.5 are marked in black. Small changes in parameter magnitudes rapidly yield extremely high-entropy distributions — essentially delta functions at 0 or 1 — except in the narrow region around the black curve representing distributions with mean 0.5.

## A  Counter Intuitive Stability Properties of the Unstable Kumaraswamy

When using the unstable KS to model latent variables with SVI, the instability of the KS distribution can *paradoxically worsen as evidence increases*. Here, evidence refers to observed data that sharpens the posterior distribution and reduces uncertainty. Representing sharper, high-entropy bell-shaped KS distributions — indicative of reduced posterior uncertainty — requires extremely large $b$ values. Figure 7 illustrates this: a bell-shaped KS distribution with mean 0.5 and differential entropy $\mathcal{H} \approx \exp(2)$ necessitates $\log b = 24$, and thus $b = \exp(24)$ in the unstable KS implementation which lacks logarithmic parameterization; see Figure 3 for examples of such moderate entropy distributions with $\log b = 24$. SVI will leverage the inverse CDF and its gradient expressions (12)–(14), which critically depend on $b$ through the term $w_{b^{-1}}(u) = \log(1 - \exp(b^{-1} \log u))$. Large $b$ values will act to worsen instability by driving $\exp(b^{-1} \log u)$ closer to 1, increasing the risk of catastrophic cancellation. We believe this counter-intuitive behavior likely frustrated modelers, but is no longer an issue in the stabilized KS.

As an illustrative example, consider modeling the latent probability of heads in a Bernoulli coin-flipping experiment using a KS distribution as the variational posterior, where the true probability of heads is 0.5. With a uniform prior and a small number of observed flips, the posterior is well-approximated by a mild-entropy, bell-shaped KS distribution, characterized by low-magnitude parameters $a, b > 1$. In this regime, $b^{-1}$ remains sufficiently far from zero, minimizing the risk of catastrophic cancellation in the term $1 - \exp(b^{-1} \log u)$, as $\exp(b^{-1} \log u)$ stays safely away from 1. However, as the number of observed flips increases, the posterior sharpens to reflect reduced uncertainty, demanding larger values of $b$ to represent the corresponding higher entropy KS distribution. This drives $\exp(b^{-1} \log u)$ closer to 1, increasing the risk of catastrophic cancellation and numerical instability.

## B  VAE Architectural and Training Choices

The following is almost identical to that used in (Loaiza-Ganem & Cunningham, 2019), but provided here for completeness. For both experiments (MNIST and CIFAR-10) we use a learning rate of 0.001, batch size of 500, and optimize with Adam for 200 epochs.

Table 4: VAE on MNIST and CIFAR-10.

| Prior | $q_\phi(\boldsymbol{z}|\boldsymbol{x})$ | $p_\theta(\boldsymbol{x}|\boldsymbol{z})$ | MNIST | | CIFAR-10 | |
|---|---|---|---|---|---|---|
| | | | LL | Acc. | LL | Acc. |
| $\mathcal{N}_{(0,1)}$ | $\mathcal{N}$ | $\mathcal{CB}$ | $\mathbf{1860 \pm 15}$ | 97.3 | $1191 \pm 907$ | 37.9 |
| $\mathrm{U}_{(0,1)}$ | KS | $\mathcal{CB}$ | $1855 \pm 14$ | 97.4 | $\mathbf{1194 \pm 914}$ | $\mathbf{41.5}$ |
| $\mathrm{U}_{(0,1)}$ | Beta | $\mathcal{CB}$ | $1856 \pm 15$ | $\mathbf{97.5}$ | $1189 \pm 914$ | 40.3 |
| $\mathcal{N}_{(0,1)}$ | $\mathcal{N}$ | Beta | $\mathbf{4209 \pm 323}$ | $\mathbf{92.1}$ | $\mathbf{3648 \pm 1210}$ | 48.5 |
| $\mathrm{U}_{(0,1)}$ | KS | Beta | $4160 \pm 331$ | 91.3 | $3579 \pm 1154$ | $\mathbf{50.1}$ |
| $\mathrm{U}_{(0,1)}$ | Beta | Beta | $4198 \pm 126$ | 90.1 | N/A | N/A |
| $\mathcal{N}_{(0,1)}$ | $\mathcal{N}$ | KS | $3393 \pm 44$ | 96.4 | $1875 \pm 890$ | 47.1 |
| $\mathrm{U}_{(0,1)}$ | KS | KS | $3401 \pm 32$ | 96.8 | $\mathbf{1939 \pm 889}$ | $\mathbf{48.8}$ |
| $\mathrm{U}_{(0,1)}$ | Beta | KS | $\mathbf{3405 \pm 40}$ | $\mathbf{97.1}$ | N/A | N/A |

**Enforcing positive variational parameters.**

- *Gaussian.* When the variational posterior is Normal, the output layer of the encoder uses a softplus nonlinearity for the positive standard deviation.

- *KS.* As we parameterize the KS by unconstrained log values, any required exponentiation occurs internally, so we require no nonlinearity on the output of the encoder.

- *Beta.* The core software libraries do not implement the Beta distribution's reparameterized sampling with unconstrained log parameter values, so we use an exponential nonlinearity on the output of the encoder to enforce positivity. A softplus nonlinearity was attempted which was found to be less stable likely due to the model seeing very large latent parameter values, which is more stably accessible via an exp.

**Enforcing positive likelihood parameters.**

- *$\mathcal{CB}$.* When the likelihood is a $\mathcal{CB}$, the output of the decoder has a sigmoid non-linearity to enforce its parameter $\lambda \in (0, 1)$.

- *KS.* As we parameterize the KS by unconstrained log values, any required exponentiation occurs internally, so we require no further transformation on the output of the decoder.

- *Beta.* The core software libraries do not implement the Beta distribution's log-pdf with unconstrained log parameter values, so we use a softplus nonlinearity on the output of the decoder to enforce positivity. An exponential nonlinearity was attempted which was found to be less stable.

**Data augmentation for $(0, 1)$ likelihood functions.** The $\mathcal{CB}$ has support $[0, 1]$ and handles data on the support boundaries without issue. When the likelihood function is a Beta or KS, which have support $(0, 1)$, we clamp pixel intensities to $\left[\frac{1}{2 \times 255}, 1 - \frac{1}{2 \times 255}\right]$ to prevent non-finite gradient values.

For all our MNIST experiments we use a latent dimension of $D = 20$, an encoder with two hidden layers with 500 units each, with leaky-ReLU non-linearities, followed by a dropout layer (with parameter 0.9). The decoder also has two hidden layers with 500 units, leaky-ReLU non-linearities and dropout. For all our CIFAR-10 experiments we use a latent dimension of $D = 40$, an encoder with four convolutional layers, followed by two fully connected ones. The convolutions have respectively, 3, 32, 32 and 32 features, kernel size 2, 2, 3 and 3, strides 1, 2, 1, 1 and are followed by leaky-ReLU non-linearities. The fully connected hidden layer has 128 units and a leaky-ReLU non linearity. The decoder has an analogous "reversed" architecture.

**Expanded experimental results.**

Table 4 includes data from the image VAE experiments in Table 2 from Section 4.1, but now including standard deviations of LL values across test samples.

## C   Kumaraswamy-Beta KL Divergence

The KL divergence between the Kumaraswamy distribution $q(v)$ with parameters $a, b$ and the Beta distribution $p(v)$ with parameters $\alpha, \beta$ is given by

$$
\mathbb{E}_q\left[\log\frac{q(v)}{p(v)}\right] = \frac{a-\alpha}{a}\left(-\gamma - \Psi(b) - \frac{1}{b}\right) + \log ab + \log\mathcal{B}(\alpha, \beta) - \frac{b-1}{b}
$$
$$
+ (\beta - 1)b\sum_{m=1}^{\infty}\frac{1}{m+ab}\mathcal{B}\left(\frac{m}{a}, b\right)
$$

where $\gamma$ is Euler's constant, $\Psi(\cdot)$ is the Digamma function, and $\mathcal{B}(\cdot)$ is the Beta function. The infinite sum in the KL divergence arises from the Taylor expansion required to represent $\mathbb{E}_q[\log(1-v_k)]$; it is generally well approximated by the first few terms.

## D   Kumaraswamy Moments

The Kumaraswamy distribution's $n$-th moment is expressed as:

$$
m_n = \frac{b\Gamma(1+n/a)\Gamma(b)}{\Gamma(1+b+n/a)} = bB(1+n/a, b).
$$

Here, $B(\cdot)$ represents the Beta function, and $\Gamma(\cdot)$ denotes the Gamma function. Using these raw moments, one can compute the variance, skewness, and excess kurtosis. As an example, the variance is given by $\sigma^2 = m_2 - m_1^2$.

## E   VBE Modified ELBO Derivation

Let $\boldsymbol{X} = [\boldsymbol{x}_1, \ldots, \boldsymbol{x}_K]$ be a matrix where the $k$-th column corresponds to the context feature $\boldsymbol{x}_k$. Assuming independence between arms and within-arm rewards, the data likelihood can be factorized as $p(\mathcal{D} \mid \boldsymbol{z}) = \prod_{(\boldsymbol{x}_a, a, r) \in \mathcal{D}} p(r \mid z_a)$. We adopt a fully factorized variational posterior of the form $q_{\boldsymbol{\phi}}(\boldsymbol{z} \mid \boldsymbol{X}) = \prod_{k=1}^K q_{\boldsymbol{\phi}}(z_k \mid \boldsymbol{x}_k)$. We set the conditional prior as a uniform distribution: $p_{\boldsymbol{\theta}}(\boldsymbol{z} \mid \boldsymbol{X}) = U_{(0,1)}^K$. Recall that $\mathcal{K}_t \subset \{1, \ldots, K\}$ represents the subset of arms that have been pulled, and thus for which we have reward data. The modified ELBO is derived as follows:

$$
\begin{aligned}
\mathcal{L}_{\beta_{\mathrm{KL}}, t}(\mathcal{D}, \boldsymbol{\theta}, \boldsymbol{\phi}) &= \mathbb{E}_{q_{\boldsymbol{\phi}}(\boldsymbol{z}|\boldsymbol{X})}[\log p(\mathcal{D} \mid \boldsymbol{z})] - \beta_{\mathrm{KL}} D_{\mathrm{KL}}\left(q_{\boldsymbol{\phi}}(\boldsymbol{z} \mid \boldsymbol{X}) \,\|\, p_{\boldsymbol{\theta}}(\boldsymbol{z} \mid \boldsymbol{X})\right) \\
&= \mathbb{E}_{q_{\boldsymbol{\phi}}(\boldsymbol{z}|\boldsymbol{X})}[\log p(\mathcal{D} \mid \boldsymbol{z})] + \beta_{\mathrm{KL}}\mathcal{H}\left[q_{\boldsymbol{\phi}}(\boldsymbol{z} \mid \boldsymbol{X})\right], \quad p_{\boldsymbol{\theta}}(\boldsymbol{z} \mid \boldsymbol{X}) = U_{(0,1)}^K \quad \text{(by design)} \\
&= \mathbb{E}_{q_{\boldsymbol{\phi}}(\boldsymbol{z}|\boldsymbol{X})}[\log p(\mathcal{D} \mid \boldsymbol{z})] + \beta_{\mathrm{KL}}\sum_{k \in \mathcal{K}_t}\mathcal{H}\left[q_{\boldsymbol{\phi}}(z_a \mid \boldsymbol{x}_a)\right] \\
&= \mathbb{E}_{q_{\boldsymbol{\phi}}(\boldsymbol{z}|\boldsymbol{x})}\left[\sum_{(\boldsymbol{x}_a, a, r) \in \mathcal{D}}\log p(r \mid z_a)\right] + \beta_{\mathrm{KL}}\sum_{k \in \mathcal{K}_t}\mathcal{H}\left[q_{\boldsymbol{\phi}}(z_a \mid \boldsymbol{x}_a)\right] \\
&\approx \sum_{(\boldsymbol{x}_a, a, r) \in \mathcal{D}}\log p(r \mid \tilde{z}_a) + \beta_{\mathrm{KL}}\sum_{k \in \mathcal{K}_t}\mathcal{H}\left[q_{\boldsymbol{\phi}}(z_a \mid \boldsymbol{x}_a)\right], \quad \tilde{z}_a \sim q_{\boldsymbol{\phi}}(z_a \mid \boldsymbol{x}_a)
\end{aligned}
$$

where in the final step, we use a single sample approximation of the expectation.

## F   VBE Prior Knowledge Incorporation

A key advantage of the VBE framework is its capacity to integrate prior knowledge about the latent variable $\boldsymbol{z}$ directly into the model. Traditional approaches often impose priors on the weights or parameters of neural networks or logistic regression models — parameters that are difficult to interpret and for which prior information is scarce. In contrast, VBEs explicitly model the latent mean rewards, making it natural to

incorporate prior information where it exists. In many multi-armed bandit applications, such as personalized recommendation systems or content curation, practitioners frequently have prior information on the probability of user engagement (e.g., the likelihood of a click or purchase). Instead of embedding this knowledge indirectly in network weights, VBEs allow one to encode it directly through the prior $p_{\boldsymbol{\theta}}(\boldsymbol{z}_k \mid \boldsymbol{x}_k)$. For example, suppose historical data or domain expertise indicates that a particular arm generally yields low rewards. In such a case, one can specify a Beta prior, $p_{\boldsymbol{\theta}}(\boldsymbol{z}_k \mid \boldsymbol{x}_k) = \mathrm{Beta}(a_k, b_k)$, with parameters chosen to concentrate the probability mass on lower values. This approach is particularly useful for large-scale commercial platforms like Amazon.com or Walmart. Users can be segmented by demographics such as age, gender, and geographic region, while items can be segmented into categories such as baby care products (e.g., diapers, wipes), power tools (e.g., circular saws, drills, screw guns), and gardening supplies. The Cartesian product of these user and item segments yields a set of user–item segments. We may not hold strong prior beliefs over the majority of the user-item segments, and so assigning the user-item segment a a uniform prior is appropriate. Conversely, for segments where robust prior information exists — for example, females aged 35–45 in New York City purchasing baby care products — a Beta prior with parameters biased toward higher probabilities is justified.

When ample historical data is available, the prior can be *learned* by including $\boldsymbol{\theta}$ in the KL divergence term of the ELBO, rather than the uniform distribution used in Section 4.2. By parameterizing $p_{\boldsymbol{\theta}}(\boldsymbol{z}_k \mid \boldsymbol{x}_k)$ using a neural network or a conditional normalizing flow for enhanced expressiveness, gradients can effectively propagate to $\boldsymbol{\theta}$.

## G   Additional Results for Variational Link Prediction with GNNs

For clarity, we present Table 5, which provides the exact numerical values corresponding to the results shown in Figure 6.

Table 5: Numerical results corresponding to Figure 6. The values represent mean $\pm$ standard deviation. Higher values are better for AUC and $\rho$, while lower values are better for Brier Score. The best value for each dataset-metric pair is highlighted in **bold**.

| Dataset | Model | AUC ↑ | Brier Score ↓ | $\rho$ ↑ |
|---|---|---|---|---|
| Cora | VEE-KS | **0.897** $\pm$ 0.015 | **0.082** $\pm$ 0.010 | **0.440** $\pm$ 0.045 |
| | VEE-Beta | 0.857 $\pm$ 0.023 | 0.117 $\pm$ 0.014 | 0.340 $\pm$ 0.021 |
| | VEE-tanh$_{\mathcal{N}}$ | 0.853 $\pm$ 0.066 | 0.115 $\pm$ 0.045 | 0.440 $\pm$ 0.043 |
| | Base-GNN | 0.870 $\pm$ 0.012 | 0.115 $\pm$ 0.045 | 0.302 $\pm$ 0.047 |
| Citeseer | VEE-KS | **0.851** $\pm$ 0.010 | **0.104** $\pm$ 0.008 | **0.304** $\pm$ 0.035 |
| | VEE-Beta | 0.789 $\pm$ 0.021 | 0.140 $\pm$ 0.018 | 0.124 $\pm$ 0.045 |
| | VEE-tanh$_{\mathcal{N}}$ | 0.850 $\pm$ 0.015 | 0.111 $\pm$ 0.004 | 0.288 $\pm$ 0.053 |
| | Base-GNN | 0.815 $\pm$ 0.006 | 0.110 $\pm$ 0.004 | 0.132 $\pm$ 0.025 |
| Pubmed | VEE-KS | **0.970** $\pm$ 0.004 | **0.061** $\pm$ 0.004 | **0.654** $\pm$ 0.015 |
| | VEE-Beta | 0.966 $\pm$ 0.002 | 0.069 $\pm$ 0.002 | 0.590 $\pm$ 0.014 |
| | VEE-tanh$_{\mathcal{N}}$ | 0.935 $\pm$ 0.030 | 0.120 $\pm$ 0.042 | 0.623 $\pm$ 0.015 |
| | Base-GNN | 0.970 $\pm$ 0.003 | 0.121 $\pm$ 0.042 | 0.590 $\pm$ 0.019 |

## H   Trade-offs in Alternative Variational Approximations

The KS distribution is a compelling choice for modeling bounded latent variables, as it satisfies the key desiderata outlined in Section 1, including support for the reparameterization trick, efficient sampling, expressiveness, and simple distribution-related functions. However, when a KS distribution is insufficiently flexible to approximate the true posterior, alternatives such as normalizing flows and copula-based variational inference (Copula VI) can offer greater expressivity, but with significant trade-offs.

Normalizing flows (Rezende & Mohamed, 2015) transform a simple base distribution through a sequence of invertible mappings, enabling rich, multimodal approximations. However, they introduce additional computational overhead, can be sensitive to hyperparameters, and may suffer from numerical instabilities. Copula VI (Tran et al., 2015) instead models dependencies between latent variables while maintaining flexible marginal distributions, but this added structure increases computational cost and can lead to higher-variance gradient estimates.

