# OpenReview forum: "Stabilizing the Kumaraswamy Distribution"
_TMLR — Accepted by TMLR_

### Review · Reviewer_Xp6w · 2025-02-08

**Summary Of Contributions:**

The paper works on the Kumaraswamy (KS henceforth) distribution, a distribution with support bounded between 0 and 1, in the context of probabilistic modeling and inference. The main contributions involve introducing an unconstrained parameterization that enables the use of the KS distribution on a range of applications. While it had been used before, numerical tricks, such as clipping, had been used for numerical stability, which sometimes could result in deteriorated performance. The parameterization used in this work appears to address these issues.

**Audience:**

Yes

**Claims And Evidence:**

Yes

**Requested Changes:**

I would be interested in seeing marginal likelihood estimates for the experiments in section 4.1.

**Strengths And Weaknesses:**

Overall I think the paper has a nice contribution of proposing a method to use the KS distribution in a numerical stable way. The proposal is shown to perform significantly better than, for instance, PyTorch’s implementation.

They show across a range of small scale experiments that the proposed approach works better than existing alternatives for the KS distribution, and that it achieves competitive performance across a range of small-scale tasks. I’d consider this last point to be a weakness in this work; the empirical evaluation, while covering a range of tasks, is all on fairly simple/small models.

A few comments / questions.

(1) Why is being able to evaluate the CDF in closed form emphasized as one of the main motivations? In the context of probabilistic inference my impression is that as long as you can evaluate the PDF, and reparameterize (low variance gradients), it should be fine?
Related, closed form entropy and KL divergences against other families are also nice to have, in the context of inference this is not required (acknowledged in this work), and using empirical estimates instead of closed form can lead to better results. See, for instance, [1].

(2) VAE experiments in 4.1 (CIFAR and MNIST).

(2.1) The section states “we evaluate using a single sample approximation of the test ELBO”. Why was this chose made? The ELBO is an expectation, would it not make sense to use more samples to get better estimates? The ELBO is later on used to compare methods.

(2.2) While table 3 shows better results for the KS likelihood (vs Beta likelihood), table 2 shows that the Beta likelihood yields better ELBOs. One way to compare methods, and that’s ultimately the goal, is the marginal likelihood of the data (the ELBO is a lower bound of that, but if we could optimize the marginal likelihood we would). Could such higher ELBOs be interpreted as the Beta likelihood fitting the data better? Ideally such a comparison should be done by estimating marginal likelihoods. I think it would be a nice addition to the results in Section 4.1.

(2.3) Also, how meaningful is comparing ELBOs for different models? For instance, when the prior changes from a uniform to a Gaussian. (Table 2, for each choice for the likelihood function, compares models with different priors.) As mentioned above, I believe a potentially more meaningful way to do this involves the marginal likelihood.

Clarification suggestion for Section 4.2. I’m not very familiar with contextual bandits. While I don’t expect a deep introduction to the topic, a brief one is useful. While the work has a short intro, I think it can be improved a bit. A few suggestions (mostly restructuring a few things):

(a) The first paragraph introduces the posterior over latent variables z, but the model was not introduced? So far the discussion only mentioned x_k, r_k, v_k.

(b) How are v_k and x_k connected? Could it be assumed v_k is a function of x_k? This is actually explained later on, but a few words earlier could help.

(c) Related to (a), the second paragraph describes the latents z_k and the distribution q(z|x), and then the likelihood p(r | v). So at this point it is unclear how the latents z fit in into the model. I think starting with the full model, and the posterior being modeled, would be useful. Again, this became clear as I kept on reading, but restructuring could help avoid these questions in the first few paragraphs.

Possible typo? Section 4.1 states “ we encode the test data x+n, and compute the mean E[q(z | x)], and …” I think the expectation should be removed, as you care about the mean of the distribution q.

[1] “Sticking the Landing: Simple, Lower-Variance Gradient Estimators for Variational Inference”. By Geoffrey Roeder, Yuhuai Wu, David Duvenaud.

---

> ### Author Response · Authors · 2025-02-23
> **Authors’ response to Reviewer Xp6w**
>
> Thanks for your time and effort in reviewing our manuscript, and for recognizing the novel contributions of our work as well as the related improvements of our proposal relative to PyTorch’s KS functionality. We appreciate your detailed suggestions, which have helped us improve both the clarity and rigor of our presentation. Point-by-point responses to your comments and associated requests for changes follow. We strive to improve our paper and will be happy to continue the discussion if any outstanding issues remain.
>
> &nbsp;
>
> **(1) On the importance of closed form CDF.**
>
> This point is well taken. We agree that for probabilistic inference, the ability to evaluate the PDF and reparameterize samples is most critical. However, closed-form CDF expressions are valuable in other settings, e.g., copulas, which further broadens the applicability of the KS distribution. Motivating comments along these lines have been included in the revised manuscript; please check the discussion surrounding Table 1.
>
> Since the CDF expressions suffer from numerical issues of the same flavor as the PDF and inverse CDF (see Section 3), we believe it naturally fits into the work. That said, we are open to de-emphasizing this motivation if you believe it would lead to an improved, more focused paper.
>
> &nbsp;
>
> **(2) VAE experiments in 4.1 (CIFAR and MNIST)**
>
> &nbsp;
>
> *2.1: Choice of Single-Sample ELBO Approximation*
>
> Thanks for this question. The sole purpose of this experiment is to provide evidence toward the stabilization of the KS (which we now spell out in the opening of Section 4.1, Discussion of results). We believe the ability of our stabilized KS to successfully converge in training and provide _comparable_ performance to other similar models (measured holistically using K-NN Acc., visualized test reconstructions, and ELBO approximations) provides such evidence.
>
> &nbsp;
>
> *2.2-2.3 Comparing Using Marginal Likelihood*
>
> Point well taken. Following your suggestion, we now report a marginal log likelihood (LL) approximation in Table 2 instead of the single-sample ELBO approximation; LL standard error results are shown in Table 4 of Appendix B. We employ the commonly used importance weighted estimator introduced in [1]. The trends remain the same: the models work comparably, with the ‘best’ model within each likelihood class typically having only marginally higher LL values.
>
> &nbsp;
>
> **Improvements to the Contextual Bandits (VBE) Section**
>
> We agree that the presentation in Section 4.2 could be revamped to improve clarity. Thanks for the detailed feedback. Following your suggestions, in the revised manuscript we have reorganized the VBE material to first formally introduce the model and then describe the variational approximation and learning procedure. We have also updated the notation (e.g., replacing $\mathbf{v}$ with the standard $\boldsymbol{\mu}$ for the unknown mean rewards) to align more closely with established MAB literature. We made similar changes to Section 4.3 to clarify the VEE presentation.
>
> It is our understanding that all the questions you raised in (a)-(c) were eventually answered, even in the original version of Section 4.2 (where the exposition was suboptimal). If there are any lingering issues you would like us to clarify, please let us know and we will take care of it in a subsequent revision.
>
> &nbsp;
>
> *Typo in Section 4.1*
>
> Good catch! This has been corrected in the revised manuscript.
>
> &nbsp;
>
>
> Thanks again for your review.
>
> &nbsp;
>
> [1] ”Importance Weighted Autoencoders”, By Yuri Burda, Roger Grosse & Ruslan Salakhutdinov

---

### Review · Reviewer_V8yS · 2025-02-11

**Summary Of Contributions:**

The paper provides detailed discussion on the KS distribution, advocating it as a flexible and efficient candidate for modelling random variables with bounded supports. Previously numerical instability has limited the use of KS and in this work the issues are resolved, by explaining a new parameterisation that makes the necessary computational elements stable. Use of KS, instead of other alternatives, is demonstrated in three concrete examples: VAE, multi-armed bandits and graph VAEs.

**Audience:**

Yes

**Broader Impact Concerns:**

None. The paper presents purely technical contribution for improving stability of basic learning algorithms.

**Claims And Evidence:**

No

**Requested Changes:**

The list below includes both minor and major comments, in somewhat random order. My overall evaluation saying the claims are not supported by evidence relates only to points 3 and 4 (and the clarifying remark in 5), whereas the other points are more just recommendations on how to improve the paper. If points 3 and 4 are properly addressed, I would definitely support publishing the paper.

1) I recommend providing the expressions for entropy, KL, moments etc for KS in Appendix, to increase the value of this paper as a stand-alone reference.

2) I think the claim of expressiveness of KS is a bit exaggerated. While it is more expressive than some of the candidates, is is still limited to the four canonical types, with no support for e.g. multimodal distributions over bounded intervals. Brief discussion on still more flexible alternatives, e.g. copulas or normalising flows (that naturally have their own clear limitations), would be useful.

3) You emphasize difference between explicit and implicit reparameterization on page 4, but do not actually show any empirical or theoretical evidence on what is the practical effect. You cite Kingma and Welling (2014) and Jang et al. (2017) with a general statement about positive qualities of explicit reparameterization, which is quite odd given that they predate Figurnov et al. (2018) that presented the basic idea of implicit reparameterization. I would strongly advice being more explicit about what is lost when resorting to implicit reparameterization (in terms of stability, computational efficiency, or whatever). Otherwise the claims related to that are not properly supported. In brief: Does it really matter if we need to use implicit reparameterization? If yes, how?

4) The role of VBE and VEE needs to be clarified. Either make them properly clear contributions of this paper or reduce their role.
For the first choice, I would start by mentioning them already in the title and abstract, but additionally you would need to extend the literature and empirical experimentation so that they are properly contrasted with the literature. For the second choice, which I strongly recommend, re-writing selected parts is sufficient. All three experiments could be communicated in similar manner, as examples of how KS can be used in interesting models and as demonstrations of how it is better than using the alternatives (beta etc) as parts of the same model. You can, and should, still mention that the specific model variant is new, but you would not emphasize their originality and would refrain making claims on the performance relative to the current state-of-the-art models.
5) Concrete examples of how VBE (Section 4.2) falls short of properly communicating a novel method, to clarify the above remark. Similar arguments, though to a lesser extent, could be raised about Section 4.3:
    - No references to any multi-armed bandit work published after 2017 in the leading paragraph of Section 4.2.
    - Only one data set and one comparison method (Xu et al. (2012)) and the data set is a specific construction with quite arbitrary elements ("raise to power of 5"). If you want to make claims on improved performance in MAB problems you need to follow standard experimentation protocols, including the data, the evaluation metrics and the baselines.
    - The VBE advantages (prior knowledge incorporation, interpretability) are general for any Bayesian model for this, not related to KS specifically, and the last point (simplicity) is questionable -- you still require specification of the NN encoder
    - The novelty of the proposed method is questionable; to me it looks like exactly standard MAB that just uses a variational approximation in Thompson sampling

5) Is log1mexp as such a contribution of this work or from Mächler (2012)? Clarify the writing.

6) $\mathcal{K}(\phi)$ is rather weird notation for accuracy -- why not just "Acc." that would fit well in the table?

7) Fig 6 is difficult to read -- please provide also the numbers in tabular format, probably in Appendix.

**Strengths And Weaknesses:**

The main message of the paper is extremely clear, and the paper has tutorial-like value in explaining how KS should be used in practice. The recommendation is well supported by empirical evidence, in three concrete scenarios. In particular the first one, a clear demonstration in a maximally simple standard setting, is easy to follow and verifies the main claim. The technical development in Section 3 is easy to read and provides enough details for e.g. all software libraries to implement KS correctly.

The main weakness is the unclear role of Sections 4.2 and 4.3 that introduce specific variants of multi-armed bandits and graph NNs building on KS. If they are intended as concrete proposals of new contributions for these highly active research areas, they should be substantially stronger in terms of related work and experimentation and there should be clear claims. On the other hand, if they are intended as quick demonstrations, similar to Section 4.1, the authors should not emphasise them specifically as contributions and perhaps should also avoid even introducing the acronyms. While there are novel elements in the solutions, they still largely appear to be minor variants of previous methods that use the KS distribution in a natural way, roughly as we would expect anyone familiar with the tools to do. To clarify, I stress that the experiments and their results are fine -- clearly showing it is better to use KS compared to beta or tanh, especially in Section 4.2 -- and my criticism is only about trying to position the specific models as contributions for the MAB and graph-NN literature.

Summary of strengths:
 - Well executed paper that provides highly practical tools for supporting use of KS in diverse applications
 - Detailed explanation of how KS should be implemented, likely to influence all key software libraries
 - Clear empirical evidence and broad illustration on how KS can be used in different kinds of applications

Summary of weaknesses:
 - VEB and VEE are introduced as novel contributions, but there is nowhere near enough space to properly evaluate them or present the related literature in detail. Consequently, the (implicit) claims made regarding these newly proposed methods is not supported properly

---

> ### Author Response · Authors · 2025-02-23
> **Authors’ response to Reviewer V8yS (Part 1)**
>
> Thanks for your time and effort in reviewing our manuscript, as well as for recognizing the work as well executed, likely to influence key software libraries, and providing clear evidence on the broad usefulness of the KS. We appreciate your constructive suggestions to improve the paper. Point-by-point responses to your comments and associated requests for changes follow. We strive to improve our paper and will be happy to continue the discussion if any outstanding issues remain.
>
>
> &nbsp;
>
>
> **Major Comments**
>
>
> Here, we aim to address your primary comments (Points 3 and 4), in order to ensure we sufficiently back our presented claims with evidence.
>
>
> &nbsp;
>
>
> *3. Impact of Explicit vs Implicit Parameterization*
>
>
> Thank you for this valuable feedback. Following your suggestion, we have updated the discussion in Section 2, “Gradient Reparameterization: Explicit and Implicit”, to make clear the practical impact of implicit vs explicit parameterization. In addition, we have replaced the earlier references with more recent work that better supports our discussion.
>
> In short, explicit parameterization can make a significant difference _in some settings_. Implicit reparameterization is more complex; for example, the common Beta implicit reparameterization method used in this work requires forward mode auto-differentiation on numerical CDF approximations, whereas the explicit reparameterization of the KS only involves implementing Eqs (12)–(14). Moreover, approximation errors in implicit methods can lead to numerical instability. For instance, in our VAE experiment on CIFAR10 (Section 4.1), neither our work nor any other study we are aware of has successfully employed a Beta variational posterior.
>
>
> &nbsp;
>
>
>
> *4. Clarifying the Roles and Originality of the VBE and VEE*
>
>
> We acknowledge the concern that the VBE and VEE were presented ambiguously — partially as demonstrations of KS’s broad applicability and partially as novel architectures capable of state-of-the-art performance. Based on your feedback, we have recalibrated and softened the claims regarding the latter.
> Specifically:
> - In Section 1 (Introduction, contribution bullets), we have adjusted the VBE contributions to "addressing exploration-exploitation tradeoffs" rather than "improving exploration-exploitation tradeoffs", and similarly for the VEE in uncertainty quantification in link prediction. Similar changes are made to Section 4; please check the opening paragraph of the revised manuscript.
> - In Section 1 (Introduction, final sentence), we clarify our claims regarding the VBE and VEE and the role of the experiments with this closing sentence: “Our experiments in Sections 4.2-4.3 provide evidence toward the benefits of adopting the KS in timely application domains — enabled by our stable parameterization — without making claims on improvements over state-of-the-art models, which future work may investigate.”
> - In Section 4.2 (Metrics and Evaluation), we have removed an explicit performance comparison claim regarding state-of-the-art methods to ensure our framing is more aligned with the role of the experiments.
>
>
> We believe these revisions better reflect the intent of the work while maintaining the significance of the empirical findings.
>
>
> &nbsp;
>
>
> _Note on the genesis of the VBE_
>
> It began as a natural extension of a standard contextual bandit model. A common approach in industry is to train a neural network that maps each arm's context $\mathbf{x_ k}$ to a Bernoulli success probability, leading to a fully observed model of the form $\prod_{k=1}^{K} \prod_{r \in \mathcal{T} _k} p(r _{i,k} \mid \mathrm{NN}(x_k; \Theta))$. The selected arm is then the one with the highest probability, with stochasticity injected (often naively) to encourage exploration (e.g., using an $\epsilon$-greedy strategy). In contrast, our VBE approach introduces an intermediate latent variable $z_k \in (0,1)$ per arm to represent its expected reward. By learning a full distribution over $z_k$ rather than a single deterministic probability, we enable the use of Thompson Sampling for a principled approach to balance exploration-exploitation tradeoffs and for the direct imposition of a prior on the expected reward to incorporate expert knowledge directly into the model. We initially assumed that such a straightforward extension would be common in the literature. However, despite an extensive search, we found no evidence of this approach. Consequently, we presented these models as novel contributions. If you are aware of such work in the bandit literature, we would appreciate your references and we would be happy to include them in the revised paper. Please see also our response to Point 5 for further discussion on the novel aspects and advantages of the VBE.

---

> > ### Author Response · Authors · 2025-02-23
> > **Authors’ response to Reviewer V8yS (Part 2)**
> >
> > **Minor Comments**
> >
> >
> > &nbsp;
> >
> >
> > *1. Including More Expressions to Make The Paper a Stand-Alone Reference*
> >
> >
> > Following your suggestion, we have included expressions for the moments of the KS distribution in Appendix D (“Kumaraswamy Moments”), while the KL divergence and entropy expressions remain in Appendix C (“Kumaraswamy-Beta KL Divergence”) and the main text (Section 2, first paragraph), respectively.
> >
> >
> > &nbsp;
> >
> >
> > *2. Expressiveness of the KS Distribution, Alternate Variational Approximations*
> >
> >
> > This point is well taken. To better calibrate the claim on expressiveness of the considered distributions, we have revised the discussion surrounding Table 1 so that it now reads “... All distributions except CB exhibit four prototypical shapes; CB is limited to two. Admittedly, none of these offer support for multimodal distributions over bounded intervals. Appendix H discusses other flexible alternatives (which have their own limitations), e.g., copulas or normalising flows.” Moreover, the “Expressiveness” row of Table 1 now reads “Relative Expressiveness”.
> >
> >
> > Please check Appendix H (“Trade-offs in Alternative Variational Approximations”) in the revised manuscript.
> >
> > &nbsp;
> >
> >
> > *5. Novelty and Advantages of the VBE*
> >
> >
> > The primary innovation of the VBE lies in its direct modeling of the expected reward, represented by the latent variable $\mathbf{z}$. In contrast, prior MAB approaches typically model the parameters $\Theta$ of the context-to-mean-reward function — often implemented as a neural network — and place a prior on these parameters. Since these network parameters are generally uninterpretable, incorporating prior beliefs about expected rewards becomes indirect and cumbersome; consequently, those methods often default to using a prior $p(\Theta)$ chosen solely for computational convenience.
> > By explicitly modeling the expected reward $\mathbf{z}$ and assigning it a prior, our approach allows for the direct and transparent incorporation of prior knowledge. Although both methods are Bayesian, we feel our formulation makes it easier to integrate meaningful prior beliefs _on the rewards_ rather than on the opaque network parameters. For a detailed discussion along these lines; see the newly added Appendix F, “VBE Prior Knowledge Incorporation.”
> > Regarding simplicity, while our approach does require specifying the neural network encoder parameters, we contend that the learning process — driven by stochastic gradient descent on the ELBO — is significantly simpler and more scalable than the matrix equation updates common in earlier work (e.g., [2] and [3]).
> >
> >
> > &nbsp;
> >
> >
> > *6. Clarification on log1mexp as a Contribution*
> >
> >
> > This is the main contribution of Mächler (2012), not our work. Following your request, this has been clarified in the revised manuscript; please check discussion immediately preceding (3).
> >
> >
> > &nbsp;
> >
> >
> > *7. Non-standard notation for Accuracy*
> >
> >
> > Thanks for this suggestion. We have updated our notation to “Acc.” in Table 2, aligning it with standard conventions. Previous notation was borrowed from [1], whose VAE experimental setup we attempt to closely follow.
> >
> >
> > &nbsp;
> >
> >
> > *8. Tabular Form of Figure 6 in the Appendix*
> >
> >
> > In response to your request, we have added a table in Appendix F (“Additional Results for Variational Link Prediction with GNNs”, Table 5) that reports the numerical values corresponding to Figure 6.
> >
> >
> > &nbsp;
> >
> >
> > Thanks again for your review.
> >
> >
> > &nbsp;
> >
> >
> > [1] “The continuous Bernoulli: fixing a pervasive error in variational autoencoders”, By Gabriel Loaiza-Ganem and John P. Cunningham
> >
> > [2]: Variational inference for the multi-armed contextual bandit, Iñigo Urteaga, Chris H. Wiggins
> >
> > [3]: VITS : Variational Inference Thompson Sampling for contextual bandits, ICML 2024, Pierre Clavier, Tom Huix, Alain Durmus

---

> > ### Comment · Reviewer_V8yS · 2025-03-04
> > **Response to both parts**
> >
> > I read through the revised manuscript and your responses, and I am overall satisfied with the modifications. I don't have any further remarks about the paper.
> >
> > The genesis of VBE story was interesting. What you propose certainly makes sense and I cannot pinpoint a specific paper doing this, but then again I am not really an expert in that area. It is maybe worth exploring further in future work. While this paper allows you to introduce the idea and I think the current claims made about the model are fine, it sounds like a paper purely focusing on this model, with full-scale empirical experimentation etc, would be interesting for the community.

---

> > > ### Author Response · Authors · 2025-03-04
> > > **Thanks**
> > >
> > > Thanks for checking our responses and the revised manuscript. We are glad to hear you are satisfied and that no longer outstanding issues remain.
> > >
> > > We agree that further pursuing VBE would be a worthwhile endeavor, and follow-up (more focused) work in this direction is a top priority within our research agenda. We appreciate your feedback and insights.

---

### Review · Reviewer_M9y9 · 2025-02-12

**Summary Of Contributions:**

The authors address the problem of alleviating numerical instabilities of the log-PDF, CDF and inverse CDF of the Kumaraswamy distribution, which is of importance in variational models with bounded latent variables. Then as illustration, they introduce a latent variable model for multi-armed bandits and graph neural networks. For the former, the goal is to improve the exploration-exploitation trade-ff while for the former the goal is to improve uncertainty quantification in link prediction tasks.

**Audience:**

Yes

**Broader Impact Concerns:**

There are no potential ethical concerns that need a broader impact statement.

**Claims And Evidence:**

Yes

**Requested Changes:**

- In the paragraph of Page 3 explaining Table 1 it is important to explain why the digamma and trigamma functions are not considered complex, and remove the redundant line about a^-1 as exp(-log a).
- Describe how the error is calculated in Figure 2 and how many terms are necessary for the log1mexp(x) approximation.
- Include error bars in Table 2.
- Include additional datasets for the multi-armed bandit experiment.
- Present other calibration metrics for the link prediction problem, e.g., Brier scores and calibration plots. The latter could be a good way to illustrate the regimes in which the KS-model perform better than the alternatives.
- The authors mention TensorFlow in the abstract, however, no other mention or examples are presented in the paper (unlike Figure 3 for PyTorch).
- In the multi-armed bandit experiment, what is the rationale for not considering a Gaussian variational distribution?

**Strengths And Weaknesses:**

Strengths: This is a well written paper. The numerical issues of the Kumaraswamy distribution are well justified and illustrated. The proposed approach is very simple and could lead to applications beyond what is considered in the paper. The utility of the Kumaraswamy distribution is illustrated on three problems, namely, variational autoencoders, contextual multi-armed bandits and graph neural networks, where the need of a bounded distribution and issues with existing approaches are demonstrated with experimental results. The experimental results show that models using the proposed stable Kumaraswamy distribution are competitive, relatively computationally efficient and may open new avenues of research using such a distribution or its underlying approximation in problems involving variational inference with bounded distributions.

Weaknesses: The novelty of the proposed approach reduces to leveraging the using the log1mexp(x) approximation of Machler (2012) in the calculation of the distribution and their gradients. The experiments though illustrate the capabilities of the proposed approximation are somewhat underwhelming. For instance:
- The experiment on MNIST favors CB as discussed by the authors and though the results on CIFAR-10 show better performance with KS-based posteriors, they seem comparable to those from a simpler Normal-Beta model.
- The multi-armed bandit experiment show more clear gains for the KS-based model, however, results are presented on a single artificial dataset, which makes the conclusions of the experiment less convincing.
- The AUC results on the link prediction problem do not show substantial gains of the proposed KS_based model relative to the simpler base-GNN. However, the correlation values do. If the goal is to quantify the quality of the predictive uncertainty, why not consider more appropriate metrics for calibration. Moreover, the authors do not provide a justification of why it is expected that their approach is better calibrated than the alternative approaches.

---

> ### Author Response · Authors · 2025-02-23
> **Authors’ response to Reviewer M9y9**
>
> We sincerely appreciate your thoughtful review and constructive feedback. We are pleased that you found our paper well-written, our contributions well-justified and illustrated, and the simplicity of our approach a strength with potential applications beyond those considered in the manuscript, including possibly new avenues of research. We value your suggestions for improvement, and we have incorporated revisions accordingly. Below, we provide responses to your comments and requested changes. We strive to improve our paper and will be happy to continue the discussion if any outstanding issues remain.
>
>
>
> &nbsp;
>
>
> *Digamma and Trigamma Function Explanation, Removal of Redundant Line*
>
>
> Thanks for these suggestions. We have revised the manuscript to clarify why the digamma and trigamma functions are not considered complex in this context. Additionally, we have removed the referenced redundant line; please check the revised discussion surrounding Table 1 in the revised manuscript.
>
>
> &nbsp;
>
>
> *Error Calculation in Figure 2, and Terms in the log1mexp(x) Approximation*
>
>
> Following your suggestion, we have updated the caption of Figure 2 to explicitly describe how the numerical error is calculated. Additionally, we have added a discussion in Section 3 [please see the paragraph preceding (3)] specifying the number of terms (typically 4) used in standard implementations of _log1p_ and _expm1_ Taylor series expansions, as these are directly leveraged in our log1mexp approximation.
>
> &nbsp;
>
>
> *Error bars in Table 2*
>
> This point is well taken. The error bars for Table 2 can be found in the revised Appendix B; please check Table 4 under ‘Expanded Experimental Results’.
>
>
> &nbsp;
>
>
> *Clarification on the Absence of TensorFlow in the Body of the Paper*
>
>
> Both PyTorch and TensorFlow provide similar implementations for the KS distribution, and neither library addresses the numerous log1mexp instabilities that we highlight. To maintain focus, we presented PyTorch as a representative example, as all our experiments were also conducted using PyTorch. The results and methodology, however, apply equally to TensorFlow. Clarifying comments along these lines have been included in the revised manuscript, please check the paragraph immediately preceding Section 3.1.
>
>
> &nbsp;
>
> *Request for Additional Datasets/Metrics*
>
> Feedback from other reviewers have prompted us to recalibrate (i.e., tone down) the claims regarding the VBE and VEE. We no longer claim superior performance to state-of-the-art. The primary role of the experiments is now to provide evidence for two key points: (i) the stability of our KS parameterization; and (ii) the broad applicability of the KS distribution across diverse settings and latent variable models. We appreciate your acknowledgment that our previous claims were well-supported by evidence and hope that our revised, weaker (but we believe still significant) claims remain compellingly supported.
>
> Following your request, we have included the Brier Score in the link-prediction experiment to support our claim of improved calibration; see Figure 6 in Section 4.3, as well as Table 5 in Appendix G (‘Additional Results for Variational Link Prediction with GNNs’). We find across datasets the KS models tend to have superior AUC, Brier Score, and uncertainty-error correlations ($\rho$). Thanks again for this valuable suggestion.
>
>
> &nbsp;
>
>
> *Rationale for Not Using a Gaussian Variational Posterior in Multi-Armed Bandit Experiments*
>
>
> The posterior is defined on $[0,1]^K$, the Cartesian product of open unit intervals, to align with the natural support of the modeled probabilities. A Gaussian variational posterior, which has support over $\mathbb{R}^K$, would introduce a support mismatch, making it unsuitable under this model. Comments along these lines have been included in the revised manuscript; please check Section 4.2, second paragraph under Experimental setup.
>
>
> &nbsp;
>
>
> *On the Comparable CIFAR-10 performance of KS-based Posteriors and the Normal-Beta Model*
>
>
> The performance of both models are indeed comparable. Here, we do not aim to introduce a more performant VAE architecture, but instead to support the aforementioned key points (i)-(ii). To clarify this point, the revised manuscript now states "Rather than introducing a more performant VAE architecture, the sole purpose of this experiment is to provide evidence toward the stabilization of the KS"; please check Section 4.1 under Discussion of results.
>
>
>
> &nbsp;
>
> Thanks again for your review.

---

> > ### Comment · Reviewer_M9y9 · 2025-03-04
> >
> > The reviewer would like to commend the authors for their response and revised version of the manuscript. Thanks for taking the time to consider the feedback and incorporating it into the revision. The edits, toned down claims and additional experiments make for a much rounded and compelling story. The novelty of the proposed approach is still limited, however, it addresses and important gap that can benefit future research efforts.

---

> > > ### Author Response · Authors · 2025-03-04
> > > **Thanks**
> > >
> > > Thanks for checking our responses and the revised manuscript. We are glad to hear you are satisfied with the revisions and that you find the paper bridges an important research gap, with tangible potential broader impacts.

---

### Author Response · Authors · 2025-02-20
**Thanks for the feedback**

We wish to thank the reviewers and the Action Editor for their constructive feedback on the original submission, which contributed to an improved revised manuscript. We are currently finalizing our point-by-point responses addressing each of the reviewers' comments and questions, as well as revising the paper to incorporate the valuable suggestions provided. Modifications to the revised manuscript will be color-coded blue to ease checking.

&nbsp;

We plan to upload our responses and the revised paper in the next day or two. We strive to improve our manuscript and will be happy to continue the discussion if any outstanding issues remain.

---

### Decision · Action_Editor_JvTy · 2025-03-13

**Recommendation:** Accept as is

**Comment:**

During the initial round of reviewers, reviewers agreed the paper was well written, they appreciated the paper's story, and felt that the approach was well supported by the empirical evidence in the paper. On the other hand, reviewers pointed out some weaknesses in the empirical evaluation and novelty of the approach.

The revision of the paper addressed most comments from the reviewers. Some reviewers felt the empirical evaluation could be still stronger, but overall agreed the paper should be accepted and that it is of interest to the TMLR community.  All reviewers agreed that the paper provides an interesting practical approach and that it addresses an important gap in the literature that could be useful for future research and applications.

**Audience:**

The paper brings a new practical light to the KS distribution, which may be of interest to users of PyTorch etc., as well as members of the audience interested in variation inference and generative modeling.

**Claims And Evidence:**

This paper studies the Kumaraswamy (KS) distribution, which supports reparameterization. The paper claims the adoption of this distribution is limited and identifies numerical stabilities associated with the pdf, cdf, and inverse cdf. In particular, the direct implementation of the KS distribution is found in existing libraries, such as PyTorch and TensorFlow. The paper proposes a stable variant of the KS distribution, and demonstrate the stability on a number of latent variable problems, such as VAEs.

The reviewers unanimously agree that the claims are well supported by the paper, especially after the recent round of revisions that have addressed issues raised regarding writing as well as provided additional experimental evidence.